# Ligand-modified nanoparticle surfaces influence CO electroreduction selectivity

Erfan Shirzadi [1,9], Qiu Jin [2,9], Ali Shayesteh Zeraati[3], Roham Dorakhan [1], Tiago J. Goncalves [2], Jehad Abed [1,4], Byoung-Hoon Lee[1], Armin Sedighian Rasouli[1], Joshua Wicks [1], Jinqiang Zhang [1], Pengfei Ou [1], Victor Boureau [5], Sungjin Park[1], Weiyan Ni [1], Geonhui Lee [1], Cong Tian [1], Debora Motta Meira [6,7], David Sinton [3], Samira Siahrostami[8] ✉ & Edward H. Sargent [1] ✉

Improving the kinetics and selectivity of $CO_2$/CO electroreduction to valuable multi-carbon products is a challenge for science and is a requirement for practical relevance. Here we develop a thiol-modified surface ligand strategy that promotes electrochemical CO-to-acetate. We explore a picture wherein nucleophilic interaction between the lone pairs of sulfur and the empty orbitals of reaction intermediates contributes to making the acetate pathway more energetically accessible. Density functional theory calculations and Raman spectroscopy suggest a mechanism where the nucleophilic interaction increases the $sp^2$ hybridization of $CO_{(ad)}$, facilitating the rate-determining step, CO* to (CHO)*. We find that the ligands stabilize the (HOOC–CH$_2$)* intermediate, a key intermediate in the acetate pathway. In-situ Raman spectroscopy shows shifts in C–O, Cu–C, and C–S vibrational frequencies that agree with a picture of surface ligand-intermediate interactions. A Faradaic efficiency of 70% is obtained on optimized thiol-capped Cu catalysts, with onset potentials 100 mV lower than in the case of reference Cu catalysts.

The CO reduction reaction (CORR) avoids the problem of carbonate crossover resulting from $CO_2$RR which leads to $CO_2$ loss in alkaline conditions[1,2]. Improving both rate and selectivity remains a priority in CORR to increase energy efficiency (EE) and decrease product separation costs[3].

Alloying and polymer coordination have been reported as strategies to enhance acetate selectivity[4–6], however, those strategies only modulate the active site characteristics and rely on other means to control the reaction environment, e.g., ionomers. Ionic liquids have been shown to enhance CORR by stabilizing reaction intermediates[7–9]. However, these have so far been successful principally in non-aqueous electrolytes, limiting their range of application thus far[10].

Using surface ligands has been previously shown to be a strategy to tune selectivity in the case of gold and silver-based catalysts[11–13]; studying ligand-based tuning is of potential interest in the case of copper, the catalyst capable of producing multi-carbon products[2,14]. Among ligands known to bind to Cu, those using the thiol end group are among the most stable[15,16], suggesting that – even under electrochemical reducing conditions – the ligand could remain chemisorbed to the catalyst surface. Thiol ligands have been previously used in CORR to affix supramolecular cages to the copper surface[17]. Unlike

[1]Department of Electrical and Computer Engineering, University of Toronto, Toronto, ON, Canada. [2]Department of Chemistry, University of Calgary, 2500 Calgary, AB, Canada. [3]Department of Mechanical and Industrial Engineering, University of Toronto, Toronto, ON, Canada. [4]Department of Materials Science and Engineering, University of Toronto, Toronto, ON, Canada. [5]Interdisciplinary Center for Electron Microscopy, École Polytechnique Fédérale de Lausanne (EPFL), 1015 Lausanne, Switzerland. [6]CLS@APS Sector 20, Advanced Photon Source, Argonne National Laboratory, 9700 S. Cass Avenue, Argonne, IL 60439, USA. [7]Canadian Light Source Inc., 44 Innovation Boulevard, Saskatoon, SK S7N 2V3, Canada. [8]Department of Chemistry, Simon Fraser University, Burnaby, BC, Canada. [9]These authors contributed equally: Erfan Shirzadi, Qiu Jin. ✉e-mail: samira_siahrostami@sfu.ca; ted.sargent@utoronto.ca

phosphine ligands where the phosphorous has limited opportunity to interact with other intermediates due to its single lone pair, sulfur may readily act as a Lewis base with neighboring intermediates and electronically (de)stabilize them[18,19].

In this work, we propose to retune the energetics of Cu catalysts in CORR, focusing on the chemical acetate. We postulate that applying a self-assembled monolayer on the Cu catalyst has the potential to alter the energetics of the rate-determining step (RDS), and that of the intermediates, leading to a change in selectivity and activity. We demonstrate a ligand modification strategy where the anchoring atom stabilizes acetate-specific intermediates electronically, while the ligand tail can be tailored to modify the local reaction environment.

## Results

### Theoretical predictions

These considerations motivated preliminary studies, using density functional theory (DFT), of alkanethiols on Cu slabs (Fig. 1) and the lateral interaction of sulfur lone pairs with $CO_{(ads)}$ (Fig. 1b). We simulate a $-SCH_2CH_3$ ligand to characterize the effect of sulfur lone pair. The calculations indicate that this interaction causes the Cu–C–O bond to bend, inducing hybridization of the C atom to shift away from $sp$ to more of a $sp^2$ character. The result is that the reduction of CO* (with $sp$ hybridization) to CHO* (with $sp^2$ hybridization) becomes exergonic in the presence of thiols, whereas on clean Cu it remains endergonic, with a total change in free energy of 0.48 eV going from pure to alkanethiolated Cu. This is the first step along the CORR pathway and the RDS to $C_{2+}$ products[20].

Prior reports have shown that increasing the alkyl chain length further has minimal influence on the calculated relative energies of intermediates[21]. We performed DFT calculations on $-S(CH_2)_2CH_3$ to $-S(CH_2)_4CH_3$ and found that – in comparison with the case of the $CH_3CH_2S–Cu$ slab – the same rate-limiting step is increased by a modest 0.05 eV (Supplementary Fig. 1a–d). This occurs since methylene groups lack functionality and are distant from the catalytic site, and thus have little impact on reaction mechanism (Supplementary Fig. 1e). However, the alkyl chain could affect the coverage, a topic further discussed below.

Next, we look at the C–C bond formation step to produce the COCHO* intermediate, an essential step in generating $C_2$ compounds[22,23]. Compared with pure Cu, C–C bond formation on the thiol-modified Cu surface is thermodynamically and kinetically more favorable, with a lower transition barrier of 0.28 eV to form COCHO* (Supplementary Fig. 2a, b). This value increases to 0.35 eV on clean Cu, indicating that thiol-modified Cu kinetically favors C–C bond formation and thus, leads to more $C_2$ compounds.

With selectivity to acetate in our view, we focused then onto the conversion of (HOC–CH₂O)* intermediate to (HOOC–CH₂)*[23], comparing with the transition instead to (HOHC–CH₂O)*, the latter favoring ethylene[23]. Our results (Fig. 1c and Supplementary Fig. 2a, b) show that, for clean Cu, the ethylene intermediate (HOHC–CH₂O)* is more stable than the acetate one, (HOOC–CH₂)*. When we modify the surface of alkanethiols, these instead favor acetate-producing (HOOC–CH₂)* by a margin of ~2.7 eV compared to the ethylene intermediate.

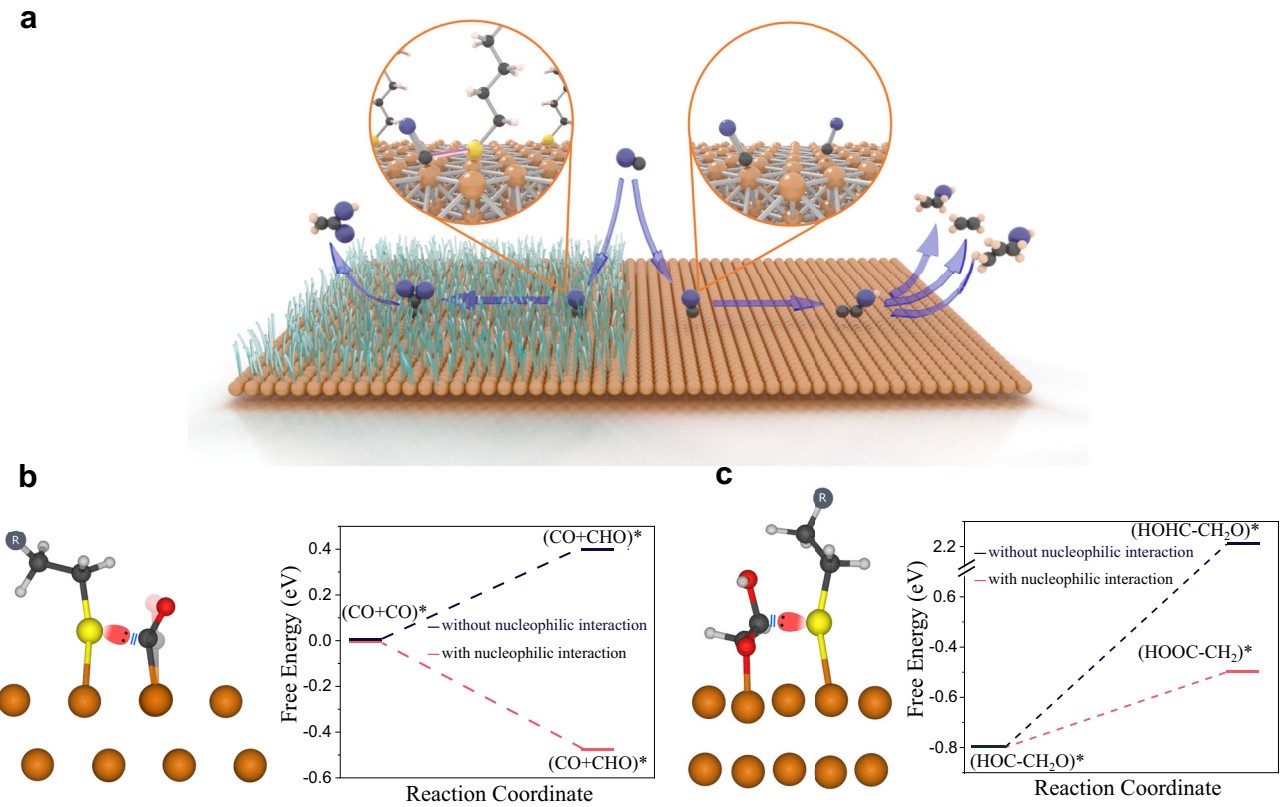

**Fig. 1 | Pathway modification via surface ligand interactions. a** A schematic showing the impact of surface ligands on the product selectivity. Geometry optimized DFT results of (**b**), the RDS for the $C_{2+}$ electrosynthesis: reduction and proton transfer to CO* on alkanethiolated slab of copper showing an interaction between C and S. This interaction leads to a higher $p$-orbital character on C and, therefore, facilitates its conversion to CHO and (**c**) Origins of selectivity toward acetate: stabilizing the (HOOC–CH₂)* intermediate drives reaction toward acetate instead of ethylene. (HOOC–CH₂)* on alkanethiolated slab of copper showing the interaction between S and C of –COOH group. (HOHC–CH₂O)* that yields ethylene, lacks this interaction and therefore is not stabilized.

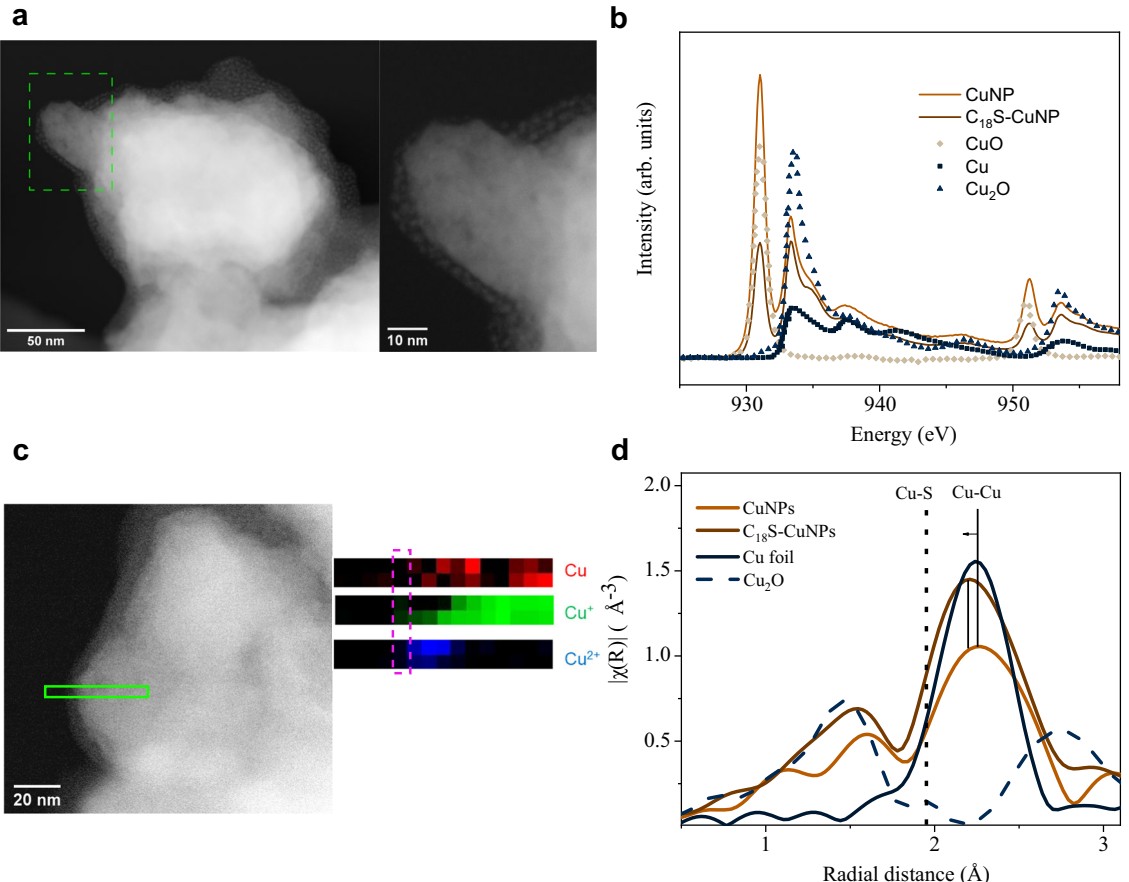

**Fig. 2 | Attachment of surface ligands to copper. a** STEM HAADF image of $C_{18}S$−CuNPs. **b** Cu $L$-edge TEY spectra of $C_{18}S$−CuNPs and CuNPs showing higher relative $Cu^I$ content in the thiolated sample. Cu, CuO, and $Cu_2O$ spectra are taken from Jiang et al[48]. **c** Color maps of the Cu oxidation state, from EELS analysis of Cu $L$-edge, across particle surface as indicated on the STEM HAADF image. **d** $R$-space EXAFS spectra of catalysts $C_{18}S$−CuNPs and CuNPs.

This is due to the sulfur lone pairs interacting with the empty orbitals of the COOH side of the acetate-producing (HOOC−CH₂)*, while the ethylene intermediate (HOHC−CH₂O)* shows negligible interaction with the sulfur and thus, is less stabilized.

To illustrate the significance of the sulfur interaction, the −SCH₂CH₃ ligand is replaced with −CH₂CH₂CH₃ (Supplementary Fig. 2c). The −CH₂CH₂CH₃ ligand did not stabilize the CORR RDS, and in fact, destabilized the formation of CHO* by 1.09 eV. We also investigated oxygen, another element in the periodic table with lone pairs similar to that of sulfur. As shown in Supplementary Fig. 3, oxygen destabilizes the CHO* intermediate by 1.02 eV and does not interact with other CORR intermediates (the distance between O of ethanolate ligand and C of CO* and CHO* is higher than their combined atomic radius). This suggests that the copper atom attached to the ligand with an oxidation state of +1 has no impact on the stabilization of the CORR intermediates, and effective interacting lone pairs are essential for this purpose. These results show the importance of atomic radius and lone pairs in sulfur's ability to interact with CORR intermediates and stabilize them.

### Structural characterizations of catalysts

Experimentally we exposed copper nanoparticles to thiol ligands dispersed in dimethylformamide (DMF) under an inert atmosphere to avoid oxidation of thiols (structures are shown in Supplementary Fig. 4). Scanning electron microscope (SEM) images show the structure of the alkanethiol-ligated copper nanoparticles (RS−CuNPs) (Supplementary Fig. 5a, b); scanning transmission electron microscopy high-angle annular dark-field (STEM HAADF) imaging reveals a thin layer of

organic thiol covering the surface of copper nanoparticles (Fig. 2a and Supplementary Fig. 5c, d). The thickness at the thinnest point is measured to be ~3 nm, close to the length of the alkyl chain of 1-octadecanethiol (2.7 nm)[24]. We don't expect long-chain alkane groups to orient vertically such that even at the thinnest point, unbound alkanethiols are absorbed via hydrophobic interaction prior to CORR. Moreover, (physisorbed) alkanethiols are expected to be removed by the alkaline electrolyte during CORR (Supplementary Fig. 5). This happens due to the acidic nature of alkanethiols reacting with KOH electrolyte.

STEM energy-dispersive X-ray spectroscopy (EDX) mapping indicates the presence of sulfur on the surface of nanoparticles, consistent with the presence of thiols on the surface of Cu nanoparticles (CuNPs) (Supplementary Figs. 6 and 7).

X-ray photoelectron spectroscopy (XPS) of sulfur on 1-octadecanethiol-ligated copper nanoparticles ($C_{18}S$−CuNPs) (Supplementary Fig. 8) shows a broad peak related to sulfur, consistent with the presence of both physisorbed (binding energy > 163 eV) and chemisorbed (binding energy < 163 eV)[25] thiols. Cu $2p_{3/2}$ XPS spectra show peaks at 933.9 and 934.5 eV for $C_{18}S$−CuNPs and CuNPs samples, respectively, indicating a CuNP surface that is fully oxidized to CuO (Supplementary Fig. 9). Additionally, CuLMM Auger spectra show a shift of 0.5 eV due to the presence of $Cu^I$ on the surface. In the RS−Cu, this contribution is assigned to the formation of $Cu^I$ upon ligand formation (Cu−S) (Supplementary Fig. 10). We ascribe this to the reaction of thiols with Cu or CuO to form $RSCu^I$ [25].

X-ray absorption spectroscopy (XAS) data collected (total electron yield (TEY) mode) for Cu $L$-edge indicates predominantly $Cu^{II}$ on

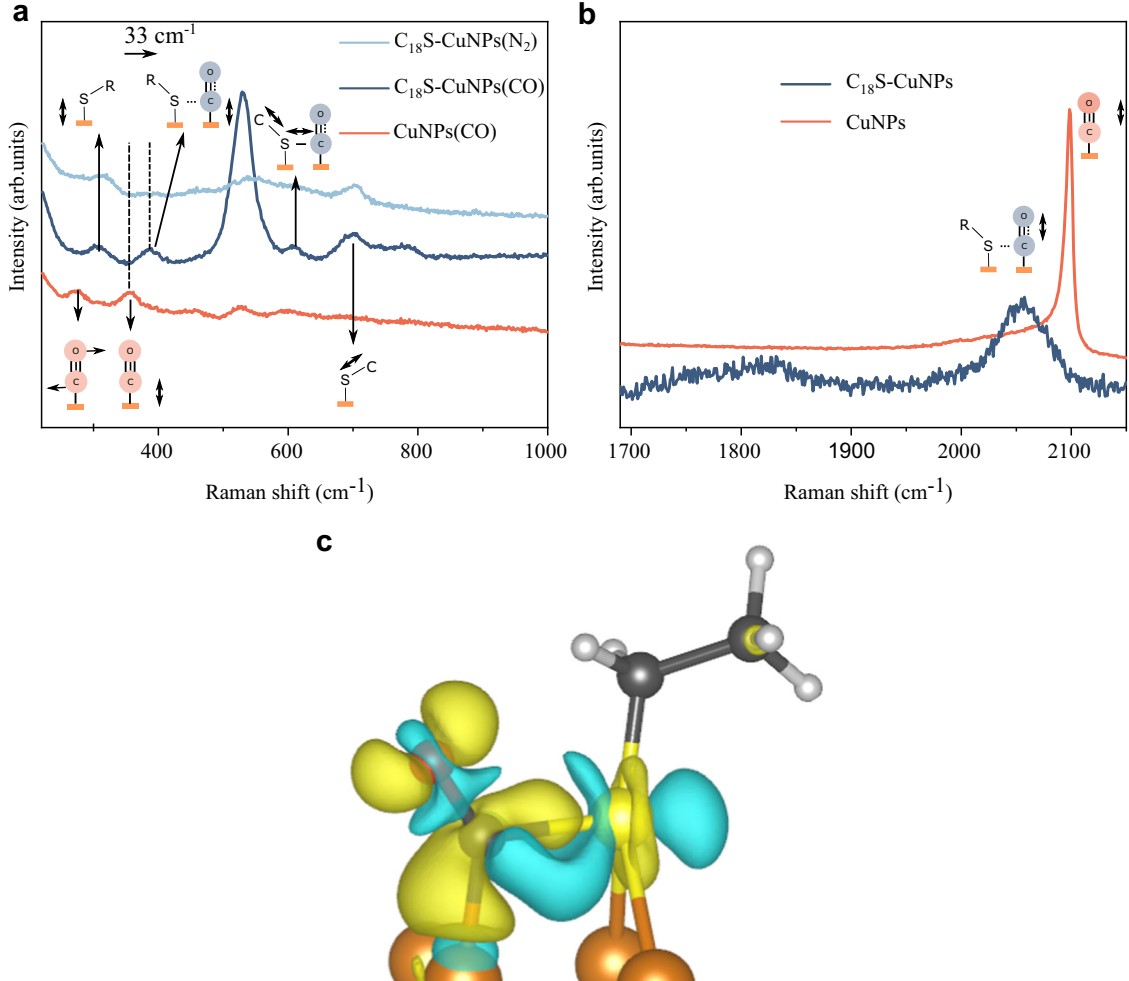

**Fig. 3 | In-situ spectroscopy and DFT results show nucleophilic interactions. a, b** In-situ Raman spectra of $C_{18}SH$–CuNPs vs. CuNPs. All the spectra are recorded in a CO or $N_2$ atmosphere (shown in the figure) assembled in a flow cell and in contact with 5 M KOH without iR correction at −1.4 V vs. Ag/AgCl. We assigned the peak at -305 cm$^{-1}$ to the Cu–S bond according to the literature[49–51]. This peak is present under an $N_2$ atmosphere, from which we conclude that it does not originate from CO-related species. The peak at 700 cm$^{-1}$ is assigned to the C–S bond stretch without the involvement of CO[52,53]. The peak at -530 cm$^{-1}$ is assigned to Cu–OH on the surface, which is based on a previous study is an indication of CORR[54]. **c** Plots, from DFT, of electron density difference for the sulfur interaction with adjacent adsorbed CO. The yellow and blue contours represent electron density accumulations and depressions, respectively. Red, O; gray, C; white, H; brown, Cu.

the surface of CuNPs. $C_{18}S$–CuNPs showed a much lower surface Cu$^{II}$ content with a much higher content of Cu$^{I}$ (Fig. 2b). Carbon $K$-edge XAS shows a sharper peak for $C1s \rightarrow \sigma^*(C-H)$ 287 eV, indicating a higher orientation order of adsorbed $C_{18}S$ ligands on copper than on $C_{18}SH$ powder (Supplementary Fig. 11)[26].

To identify the oxidation state of copper across the thickness of the CuNPs, we analyzed the fine structure of Cu-L edge from STEM electron energy-loss spectroscopy (EELS) measurements. Comparing CuNPs and RS-CuNPs, we find that RS-Cu samples are much less prone to oxidation, evident from the higher content of Cu[(0)]. The higher oxidation resistance was previously observed on self-assembled monolayers of n-alkanethiolates on copper[15]. The surface of RS–CuNPs contains mainly Cu$^{I}$ (Fig. 2c), whereas the surface of the CuNPs contains predominantly Cu$^{II}$ (CuO) (Supplementary Fig. 12).

To study the copper coordination in alkane-thiolated samples, we employed XAS analysis. As shown in Fig. 2d, when inspecting the extended X-ray absorption fine structure (EXAFS) region, one observes a shift in the radial distance related to the Cu–Cu bond compared to pure copper. The shift can be assigned to the coordination of sulfur to copper in the scattering of ejected photoelectrons. This is indicated by EXAFS simulations of alkane-thiolated copper slabs (Supplementary Fig. 13). Although XAS is predominantly a bulk analysis technique, the

high coverage of thiols on the surface and the use of the more sensitive fluorescence detector, allowed for the detection of the sulfur signal in the R-space of EXAFS. However small, this signal produced a noticeable shift in the Cu–Cu region at -2.2 Å. Since copper and sulfur have similar atomic radii and the sulfur contribution is small, we observe it as a shift in the Cu–Cu region rather than a separate peak.

## In-situ Raman spectroscopy and XAS experiments

Next, we used in-situ Raman spectroscopy to probe the interaction between sulfur lone pairs and CO (Fig. 3a). On Cu, the peak at 356 cm$^{-1}$ is assigned to the $\nu$(Cu–CO) vibrational frequency[27–29]. This frequency on the thiolated samples undergoes a blueshift to 388 cm$^{-1}$. Comparing the atop CO stretch frequencies, we detect a redshift from 2099 cm$^{-1}$ to 2055 cm$^{-1}$ in the alkanethiolated samples compared to CuNPs (Fig. 3b). Additional experiments with 2-mercaptopyridine (2-Mpy) revealed a similar thiol–CO* interaction (Supplementary Note 1).

To assist the interpretation of the Raman spectral shifts, we used DFT to investigate the difference in the electron density map for CO* in the presence vs. absence of thiols. Moreover, DFT-predicted Raman shifts were also measured. An increase in the electron density of the Cu–C bond and a reduction in the C–O bond is observed in the presence of thiols (Fig. 3c). The Raman spectra of CO* were simulated

using DFT calculations on an ethanethiol-ligated ($C_2$S–Cu) and bare Cu surface (Supplementary Fig. 14). Supplementary Movie 1 shows the vibration simulation of the Cu–S bond at 311 $cm^{-1}$ on the $C_2$S–Cu slab and Supplementary Movies 2 and 3 illustrate the vibration simulation of the Cu–CO on the clean Cu and $C_2$S–Cu slabs, respectively. The $v$(Cu–CO) vibration peak of $C_2$S–Cu slab appears at higher Raman shifts compared to the clean Cu slab. In contrast, the primary vibration related to $v$(C–O) on the $C_2$S–Cu slab is observed at lower Raman shifts (Supplementary Fig. 14c). This is in agreement with the higher Raman shift experimentally observed for $v$(Cu–CO) vibration on $C_{18}$SH–CuNPs (Fig. 3a). Simulations of Raman spectra give a shift of 620 $cm^{-1}$ for the C–S bond, in good agreement with the experimental value (-610 $cm^{-1}$) for $C_{18}$S–CuNPs. Supplementary Movie 4 shows the vibration related to C–S bond. Taken together DFT and Raman spectroscopy results, it suggests a sulfur lone pair interaction with adsorbed CO; suggestive of a route to enhance CORR performance.

It will be of interest to seek experimental evidence of the conversion of (HOC–CH2O)* intermediate to (HOOC–CH2)* and to observe, experimentally, the lone pair interaction at this step. In the present study, we were not able to witness this directly, something we attribute to the fact that the conversion of (HOC–$CH_2$O)* intermediate to (HOOC–$CH_2$)* happens after the RDS, and thus its coverage is low.

Next, we probe whether – under operating potentials, and in the presence of CO – the thiol layer will remain present on the Cu catalyst surface. In-situ XAS experiments show that sulfur atoms remain on the surface throughout the reaction (Supplementary Note 2). We detected the Cu–S bond for $C_{18}$S–CuNPs at potentials up to −1.5 V vs. reversible hydrogen electrode (RHE) via in-situ Raman study (Supplementary Fig. 15). To further assess the stability, we used DFT calculations to simulate the voltages in which alkane-thiol ligands remain bonded to the Cu surface (Supplementary Fig. 16). According to the results, $C_2$S ligands are stable up to −1.2 V vs. RHE, close to the experimental value observed (−1.5 V) for the $C_{18}$S ligand. Prior reports in $CO_2$RR have indicated the presence of Cu(I) species under pulsed conditions[30,31], however, under constant current experiments and in CORR conditions where the buffering capacity of $CO_2$–carbonate is not present, we find that CuNPs transform to their Cu(0) oxidation state. Moreover, results from DFT calculations (Supplementary Note 2) find the Cu atoms attached to alkanethiols to have a positive charge. This is reflected in the shift in XANES curves of the $C_{18}$S–CuNP compared to CuNP.

## Electrochemical CORR performance

We used a gas-diffusion electrode (GDE) to evaluate the electrochemical performance of thiol-modified NPs under current densities of ≥100 mA.$cm^{-2}$. The majority of RS–CuNPs showed the lowest HER values, a finding we assign to their hydrophobicity. As controls, we screened other ligand surface modifiers that possess hydrogen bond acceptor/donor functional groups, with all samples showing a higher HER activity (see Supplementary Table 3). However, the higher HER can be due to different coverage of thiols in these catalysts.

Under CORR, the reaction onset potential (Fig. 4a) of the thiol samples is lower by 100 mV compared to controls. To rule out surface area effects, we measured the electrochemically active surface area (ECSA) of the electrodes, finding that, in fact, the thiol samples had lower effective area than controls (Supplementary Table 4 and Supplementary Figs. 17–20). This confirms that the per-site activity of the alkanethiolated samples is indeed higher than that of CuNPs based on the onset potential. Furthermore, we measure a 120 mV/dec Tafel slope on thiol samples, suggesting the first electron transfer step to be rate-determining (Supplementary Fig. 21)[20,32,33].

Next, we studied the impact of alkane-thiol chain length (Supplementary Table 3) from ethane ($C_2$) to docosane ($C_{22}$). We did not observe spikes in hydrogen evolution reaction (HER) indicative of diffusion limitations up to current densities of 400 mA/$cm^2$ for $C_{18}$S–CuNPs and $C_{12}$S–CuNPs (see Supplementary Tables 5–7). However, the $C_5$S–CuNPs exhibited a considerable limitation in CORR current density, with HER FE values spiking to >60% at 200 mA/$cm^2$ (see Supplementary Table 5). This is attributed to the dense packing capacity of smaller thiols and their subsequent high coverage. This effect is exaggerated on even shorter $C_2$SH thiols, where HER activity was higher compared to $C_5$S–CuNPs. This may also be due to the inability of short ethyl chains to establish a proper triple-phase boundary. On higher chain length samples such as $C_{22}$S–CuNPs, an increase in HER is also observed at 200 mA/$cm^2$. This could be due to the lower ECSA of this catalyst (Supplementary Table 4). Nevertheless, all $C_2$ to $C_{22}$ alkanethiols showed better FEs for acetate than CuNPs.

The mid chain length samples $C_5$S–CuNPs to $C_{12}$S CuNPs showed a superhydrophobic surface, with little change in contact angles across samples (Supplementary Table 8). Although the $C_2$S–CuNPs also exhibited a superhydrophobic surface, we were not able to measure the receding contact angle for this sample as the surface got wet after the force was applied. This can be justified considering the surface of $C_2$S–CuNPs is only covered by a very short alkane chain which cannot repel water under pressure.

To better understand the alkane-thiol coverage, we measured ECSA for the RS–CuNPs samples (Supplementary Table 4 and Supplementary Figs. 17–20). We posit there to be an inverse relation between the ECSA and alkanethiol coverage as the alkanethiolated sites do not contribute to the ECSA measurement. Moreover, the alkane-thiols would block ions from reaching the surface and therefore reduce the ECSA further.

On the $C_2$S–CuNPs sample, the alkyl chain of only two carbons is not as effective in blocking the ions, evident from its higher ECSA (29.3 $cm^2_{ecsa}$) compared to other RS–CuNPs. The ECSA of $C_2$S–CuNPs catalyst is almost half of bare CuNPs (62 $cm^2_{ecsa}$). As shown in Supplementary Table 4, $C_{12}$S–CuNPs and $C_{18}$S–CuNPs have similar ECSA (16.5 and 15.5 $cm^2_{ecsa}$, respectively) and $C_5$S–CuNPs has a slightly lower ECSA (8.3 $cm^2_{ecsa}$), attributed to the denser packing of the shorter alkane-thiols ($C_5$ vs. $C_{12}$ and $C_{18}$). The $C_{22}$S–CuNPs had an ECSA of only 5 $cm^2_{ecsa}$, due to the long alkane chains inhibiting ion diffusion to the surface. To support these findings, we conducted inductively coupled plasma (ICP) measurements. Our ICP results revealed RS–CuNPs with shorter chain alkane thiols have higher S:Cu ratio than RS–CuNPs with longer alkanethiols (Supplementary Fig. 22). We also estimated the coverage of thiol ligands based on ICP results and calculating the percentage of surface copper atoms in 25 nm copper nanoparticles. We estimated the coverage of thiols for $C_2$, $C_5$, $C_{12}$, $C_{18}$, and $C_{22}$S–CuNPs as 85, 53, 47, 20 and 16%, respectively (Supplementary Fig. 22). This indicates that, to achieve high C–C coupling and acetate production at high current densities, we require fewer than half of copper sites to be unligated. In view of high coverage of short chain $C_2$S–CuNPs, C–C coupling is difficult, and high HER efficiencies are the result.

We explored the effect of ligand loading, finding that higher loadings inhibit HER (Fig. 4b); an effect that saturates, presumably after a near-monolayer coverage is achieved. Higher loadings of $C_{18}$SH render the surface more hydrophobic and reduce the ECSA, demonstrative of a high coverage of alkane-thiols (Supplementary Table 8). The ECSA reduction continues until a loading of 5 mg, with no significant change in ECSA at a higher loading of 15 mg (Supplementary Fig. 23). It is worth mentioning that the initial loading is not the same as the loading during the electrochemical reaction as we expect the physisorbed alkanethiols to be removed in alkaline conditions.

By optimizing the loading, coverage, and chain length of the alkane-thiols, the Faradaic efficiency (FE) toward acetate can be increased to 70% at 400 mA/$cm^2$, a near 57% increase compared to the Cu controls (Fig. 4c, d). This FE is among the highest reported which operates at low overpotentials[4] (Supplementary Table 9).

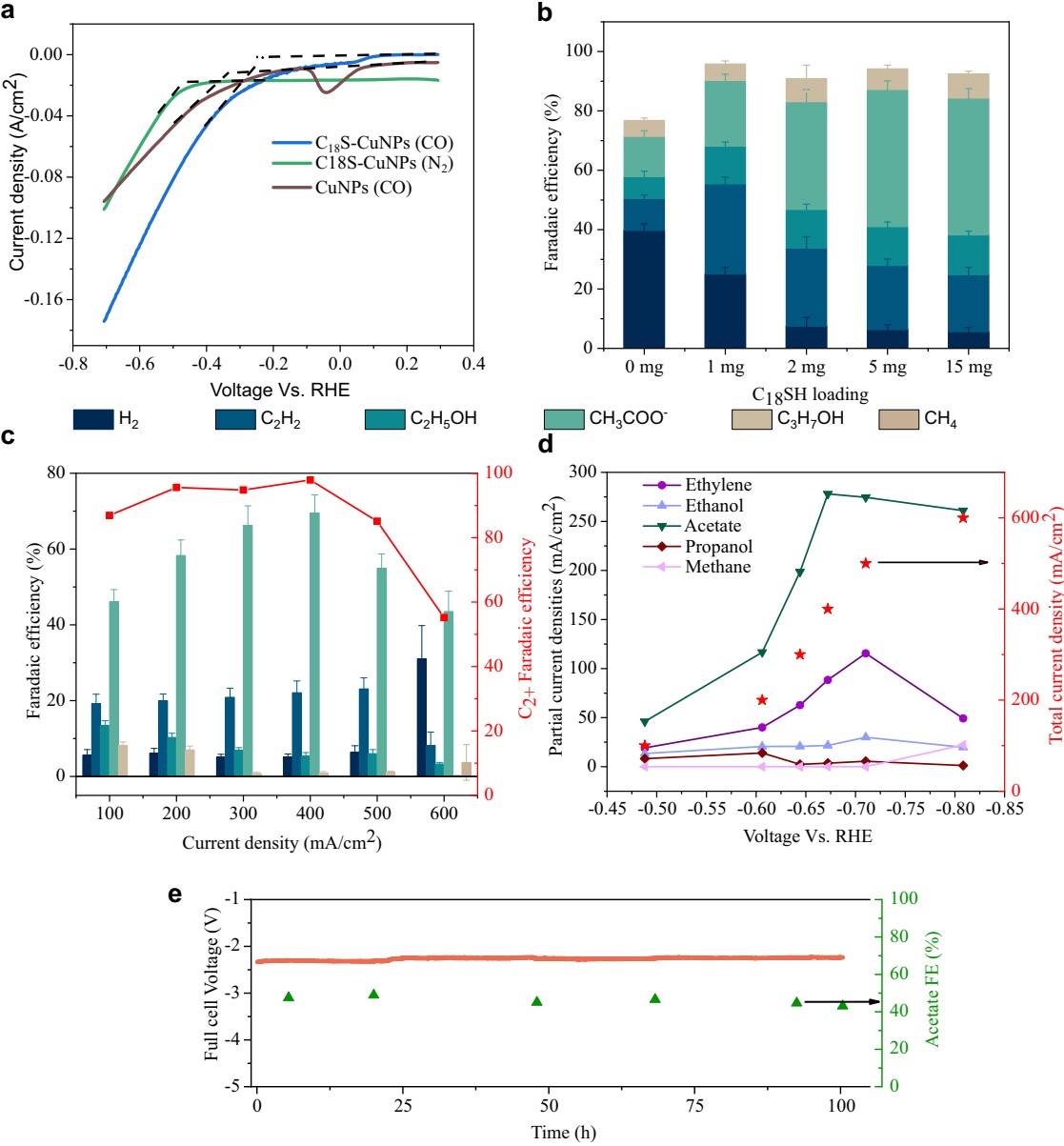

**Fig. 4 | Electrochemical performance of $C_{18}S$–CuNPs. a** Linear sweep voltammetry data of CuNPs and $C_{18}S$–CuNPs show a lower onset potential for $C_{18}S$–CuNPs. Voltages are not iR corrected. **b** Faradaic efficiency distribution of CORR products for different loadings of $C_{18}SH$ at 100 mA/cm². Data are shown as the mean ± SD, $n = 3$. **c, d** The electrochemical performance of $C_{18}S$–CuNPs at different current densities. For current densities larger than 400 mA/cm², we observed an increase in

HER and, consequently, a decrease in acetate and $C_{2+}$ FEs. This is attributed to the CO mass transfer limitations. The voltages corresponding to 100, 200, 300, and 400 mA/cm² are −0.49, −0.61, −0.64, and −0.67 V vs. RHE, respectively (iR corrected). Data are shown as the mean ± SD, $n = 3$. **e** Long-term operation test of $C_{18}S$–CuNPs in an MEA system. An $IrO_2$ anode and a solution of 1 M KOH was used as anolyte.

## Stability tests

Next, we move to demonstrate the catalyst in a full-cell system setup to characterize EE. In a zero-gap membrane-electrode assembly (MEA)[34] and using 1 M NaOH as an anolyte and an $IrO_2$ anode, we obtained 5.5 wt% sodium acetate solution, with stability over 35 h with a full-cell voltage of 2.3 V at 100 mA/cm² (Supplementary Fig. 24). Furthermore, using a similar setup and 1 M KOH anolyte and an anion exchange membrane, 100 h of stable operation were demonstrated (Fig. 4e, Supplementary Table 10) at a full cell voltage of 2.25 V and a current density of 100 mA/cm². The device provided an EE of 25% for acetate, a near 2× increase relative to the best prior reports[4,34].

The alkanethiol-covered copper nanoparticles can also be used in $CO_2RR$ as an effective way to obtain high $C_{2+}$ products[35]. Covering

CuNPs with octadecanethiols allows the copper nanoparticles to operate without needing Nafion binders and have current densities as high as 300 mA/cm² (Supplementary Table 11).

We examined whether a hydrophobic amine like oleylamine can behave similarly in CORR. However, the performance was similar to bare CuNPs suggesting that oleylamine did not remain attached to copper under our experimental conditions (Supplementary Table 12).

## Discussion

Overall, we explored two aspects of alkanethiols: the impact of available sulfur lone pairs adjacent to the surface CORR intermediates and alkyl group length. Sulfur lone pairs were directly involved in the transformation of CORR intermediates and reduced the onset

potential for CORR by -100 mV. The intermediates having empty π* are the most affected by sulfur lone pairs and they are crucial for obtaining acetate. Such lateral interaction is uncommon in molecular catalysis since the geometry of a single metal site does not allow for effective interaction.

Although the surface of copper nanoparticles in RS−CuNPs is well-covered by thiols, in the case of $C_{18}S$−CuNPs and $C_{12}S$−CuNPs, the removal of physisorbed thiols is enough to make enough room for reactants to react. CO molecules are weakly polar (with only 0.122D), and hence more soluble in apolar alkanes and organic solvents than water[36]. Thus, CO is expected to be well dissolved into alkane chains of alkanethiols. Without alkane-thiols that dissolve CO molecules, we cannot reach current densities higher as high as 100 mA/cm² and we would need a binder like Nafion (Supplementary Table 3).

On the other hand, the alkyl groups assist in the formation of triple-phase boundary, and their size mainly affects the packing. As a result of high coverage, HER rises at current densities above 100 mA/cm² (Supplementary Table 5). Nevertheless, these catalysts exhibit relatively high acetate FE, indicating they follow the same reaction mechanism. $C_2S$−CuNPs is a special case in which ethyl groups are insufficient to establish proper triple-phase boundaries and also have a high thiol coverage, leading to high HER. We conclude that choosing the proper interacting element (sulfur in our case) and optimum alkyl chain ($C_{18}$ and $C_{12}$) is crucial for obtaining a high-performance acetate electrocatalyst. Using alkanethiols brings four benefits overall: low overpotentials due to sulfur lone pair interaction, high acetate FEs and, low HER, and enhanced operating stability.

We identify a number of opportunities for future work informed by the findings herein. DFT studies would beneficially explore the full range of alkyl chain length, including those values that are optimal experimentally, $C_{12}$−$C_{18}$. Chain length could also influence the diffusion of reactants, something that could be studied using molecular dynamics.

## Methods

### Materials
KOH, alkanethiols, 2-Mpy, dimethyl sulfoxide (DMSO), deuterated water, and methanol, were used as purchased from Aldrich. In total, 25 nm copper nanoparticles were purchased from US Research Nanomaterials. Carbon monoxide (Grade 4) was purchased from Praxair and Ar (Grade 5) was purchased from Messer. Nickel foam was purchased from MTI Corporation. Carbon-based GDE (Freudenberg H23C9), anion exchange membrane (Fumasep FAA-PK-130), Nafion 212. Deionized water (18.2 MΩ) was used in all experiments.

### Synthesis of electrocatalysts
CuNPs (10 mg) were sonicated for 2 h prior to the thiolation. Then they were poured into a solution of 1, 2, 5, and 15 mg of 1-octadecanethiol in 15 ml of DMF (3.5 mM). The resulting mixture was kept under an inert atmosphere for 4 h with constant stirring. Afterward, the reaction mixture was centrifuged at -5000 × g and washed with methanol three times to remove the excess thiols. A similar procedure was employed to prepare RS−CuNPs catalysts with other thiols. 3.5 mM concentration of thiols was used for all preparations.

### Preparation of gas-diffusion electrode
The catalyst was deposited on a Freudenberg H23C9 (Fuel Cell Store) GDE using an airbrush technique, with GDE dimensions of (3 × 3 cm²) or (2 × 2 cm⁻²). The ink was composed of catalyst nanoparticles and 5 ml of methanol (Sigma Aldrich). The ink composition was chosen to obtain a 0.8 mg cm⁻² catalyst loading. The ink was vortexed for 5 min and sonicated for 30 min before being deposited on the GDE. The electrodes were kept in the glovebox before electrochemical measurements were conducted.

### Transmission electron microscopy (TEM)
The dry particles were deposited on lacey carbon thin films for scanning transmission electron microscopy (STEM) analysis, using a FEI Titan Themis[3] equipped with a high-brightness field emission gun operated at 200 kV. A probe convergence angle of 20 mrad was used, and STEM aberrations were corrected with a CEOS DCOR up to the 4th order.

HAADF STEM images were collected with a probe current of 120 pA over a range of collection angle of 50−200 mrad.

EDX spectrum images (SI) were acquired with a 800 nA-current probe scanned on the specimen with a step size of 1.3 nm and a dwell time of 50 μs, using a windowless silicon drift detector providing a solid angle of 0.7 srad. Multiple frames were summed to increase the signal-to-noise ratio of the measurement. Elemental net count maps were processed using the Kα lines of C, O, S, and Cu elements.

EELS experiments were carried out with a probe current of 120 pA and a collection angle of 47 mrad. A GIF Quantum ERS system was used to acquire dual-EELS SI with an energy channel of 0.1 eV, centered on the 0-loss (low-loss) and Cu-L edge energy (high-loss). SI of 2 × 15 pixels were acquired using a sub-pixel sampling for a total pixel time of 15 s and a pixel width of 3 nm, at the edges of the specimen particles. The full width half maximum of the 0-loss peak is 1.0 eV. The high-loss spectra were processed as follows. The energy shift was aligned with respect to the low-loss SI and the background was subtracted. The energy loss near-edge structure (ELNES) of the Cu-L line was investigated in the different SI in order to identify three reference spectra, characteristics of the different oxidation states of Cu[37]. These spectra are shown in Supplementary Fig. 12c, after Satitzky−Golay filtering. The two main features of the ELNES (L2 and L3 edges) are the sharpest for $Cu^{2+}$ and the smoothest for Cu. $Cu^{2+}$ ELNES also shows a shift of the L-edge toward lower energies. Finally, a multiple linear least squares fitting was performed on the Cu-L ELNES of the SI with the three reference spectra. The coefficient maps show the spatial distribution of the oxidation states of Cu.

### X-ray absorption spectroscopy (XAS)
Hard XAS measurements for the K-edge of Cu was carried out at 20BM beamline in the advanced photon source (APS, IL, USA) and at SXRMB beamline in the Canadian Light Source (CLS). A custom-made three-electrode flow electrochemical cell was used to collect XAS spectra in situ during CO/CO₂ reduction[38]. A Ni foam rod and Ag/AgCl electrode were used as counter electrode and reference electrode, respectively. The catalyst was prepared on one side of a GDE while the other side was facing a thin Kapton window (polyimide film with silicone adhesive) and connected electrically by Cu tape. Then, the sample was mounted in the electrochemical cell so that the backside of the sample is facing the beam.

In-situ experiments were conducted at two conditions: OCP, −2V vs. Ag/AgCl in 1 M KOH. Data was collected in fluorescence detector placed at 45°. Data post-processing and fitting were done entirely using Demeter software package[39].

Ex-situ XAS measurements for the Cu L-edge, S L-edge, C K-edge, and O K-edge were performed at the Spherical Grating Monochromator (SGM) beamline of the CLS. All samples were scanned from in 0.1 eV steps. Surface sensitive absorption spectra were recorded using TEY and partial fluorescence yield simultaneously.

### Quantification of average copper oxidation state
Based on the linear relation between the copper K-edge and oxidation state described before[40], we plot our K-edge values of our sample and normalized its oxidation state between 0 (sputtered copper under reduction) and 1 (cuprous oxide). The K-edge was measured as the value of the peak when first derivative of absorption is plotted against energy (eV).

### X-ray photoelectron spectroscopy (XPS)

Measurements were carried out using the PerkinElmer model 5600, which was outfitted with a monochromated Al Kα X-ray source emitting photons with an energy of 1486.6 eV. The samples under investigation were prepared on conductive glass substrates by drop-casting a few droplets of the ink solution.

### Electrochemical CO reduction measurements

Electrochemical rate measurements were conducted in two setups of a flow cell and MEA. In the flow cell arrangement, the catalyst-deposited GDE served as the working electrode (cathode), while nickel foam (MTI Corporation) acted as the counter electrode (anode). An anion exchange membrane (Fumasep FAA-PK-130) was utilized as separating membrane. The flow cell comprised the GDE, anion exchange membrane, and nickel anode, with 5 M KOH (Sigma Aldrich) as electrolyte circulated in both anode and cathode compartments via a peristaltic pump (McMaster-Carr). Carbon monoxide (Praxair, Grade 4.0) was directed behind the GDE using a mass flow controller (Sierra Smart-Track 100), and an Ag/AgCl electrode was used as the reference electrode in the cathodic compartment.

In the MEA setup, the catalyst-deposited GDE served as the working electrode (cathode), and an $IrO_2$−Ti felt (Sigma Aldrich) acted as the anode. A Nafion 212 (Fuel Cell Store) cation-exchange membrane was employed for anolyte and catholyte seperation. The MEA was constructed by layering the GDE, Nafion, and $IrO_2$−Ti felt, sandwiching them between the stainless steel cathode plate (with $1\,cm^2$ flow field) and the titanium anode plate (with $5\,cm^2$ flow field). A 1 M NaOH solution was circulated through the anodic flow field via a peristaltic pump, while carbon monoxide was directed into the cathodic flow field using a mass flow controller. Liquid products, containing a high concentration of sodium acetate, were collected in a cooled container (2–6 °C) from the cathode plate outlet, and after NMR analysis, the sodium acetate concentration was determined to be 5.5%.

In another MEA setup designed for stability monitoring, a 1 M KOH solution was employed in the anode, and a Sustainion X37-50 (Fuel Cell Store) served as an anion exchange membrane. This arrangement facilitated the crossover of acetate produced in the working electrode (cathode) to the anodic KOH reservoir. In all configurations, the carbon monoxide flow rate ranged from 25 to 40 standard cubic centimeters (SCCM), while the electrolyte flow rate was maintained between 10 and 15 ml min⁻¹. Electrochemical reactions were conducted using an electrochemical workstation (Autolab PGSTAT302N) connected to a current booster (Metrohm Autolab, 10 A). Electrode potentials were rescaled to the RHE reference:

$$E_{Vs\,RHE} = E_{Vs\,Ag/AgCl} + 0.197\,V + 0.059 \times pH \qquad (1)$$

Electrochemical impedance spectroscopy was conducted using an Autolab PGSTAT302 Nelectrochemical workstation to determine a cell resistance of 1.67 Ω. Subsequently, iR corrections to the potential were applied using the following equation:

$$E_{iR-free} = E_{Vs\,Ag/AgCl} - 0.85 R_{cell} i_{cell} \qquad (2)$$

where $E_{iR-free}$ represents the corrected potential at the cathode, $E_{vs\,Ag/AgCl}$ denotes the applied potential, and $i_{cell}$ stands for the total current (negative at the cathode). A correction factor of 0.85 is applied to account for the high conductance of the 5 M KOH electrolyte and its low voltage drop across the electrolyte.

The EE of the entire MEA system is derived from the subsequent equation:

$$EE_{Full-cell} = FE_{Acetate} \times \frac{E_{cell}^0}{V_{Full-cell}} \qquad (3)$$

Here, $EE_{Full-cell}$ represents the full-cell EE of the system, $FE_{Acetate}$ denotes the system's FE toward acetate, and $V_{Full-cell}$ stands for the average full cell voltage throughout the duration of the stability experiment. The standard reduction potential of the cell is derived from the standard Gibbs free energy of the reaction: $2CO + 2H_2O \rightarrow CH_3COOH + 2O_2$:

$$E_{cell}^0 = \frac{-\Delta G_{2CO+2H_2O \rightarrow Acetate+2O_2}^0}{zF} \qquad (4)$$

where $z$ equals 4, and $F$ represents the Faraday constant, which is 96485 C mol⁻¹.

### Product analysis

Gas products resulting from CORR underwent analysis by extracting 1 ml of the electrolyzer outlet gas via an air-tight syringe (Hamilton, 5 ml). This gas sample was then introduced into a gas chromatograph (Shimadzu GC-2014) equipped with both a thermal conductivity detector (TCD) and a flame ionization detector (FID). To facilitate separation of $H_2$, CO, and gaseous hydrocarbons, a molecular sieve 5A column was placed upstream from the TCD, while a Carboxen-1000 column was positioned upstream from the FID. Liquid products were quantified through nuclear magnetic resonance spectroscopy (NMR), involving the extraction of 1 ml samples of the electrolyte at various intervals during the reaction. ¹H NMR spectra of the analyte samples were acquired employing an Agilent DD2 600 spectrometer, with DMSO serving as the internal standard. To ensure precise measurement of acetate concentration, an HPLC system fitted with an HPX87H Aminex column (Bio-Rad) was calibrated specifically for acetate. The findings obtained from both the HPLC and NMR analyses exhibited an error margin within 5%. FE for CORR gas products and $H_2$ was determined using the subsequent equation:

$$FE_{i,gas} = y_{i,g} \dot{V} z_i F \frac{P_0}{RT} j_{total} \qquad (5)$$

where $y_i$ represents the volume fraction of gas product i, $V$ denotes the outlet gas flow rate in SCCM, $z_i$ signifies the number of electrons associated with producing one molecule of i, $F$ stands for the Faraday constant, $P_0$ denotes atmospheric pressure, $R$ represents the ideal gas constant, $T$ signifies the temperature, and $j_{total}$ indicates the total current density. The flow rate was manually determined using a bubble flowmeter. The FE of the liquid product was computed using the subsequent equation:

$$FE_{i,liquid} = n_{i,liquid} z_i F \frac{1}{Q_{total}} \qquad (6)$$

Where $n_i$ represents the quantity of moles of liquid product i, and $Q$ denotes the total charge passed before extracting the liquid sample.

### ICP measurement and coverage calculation

The concentration of sulfur and copper in the catalyst was measured by ICP-OES. Calibration solutions of sulfur and copper were prepared by diluting respective standards of 1000 ppm (Sigma Aldrich, *Trace-CERT*) to obtain solutions of 50, 10, 5, 1, and 0.2 ppm. Dilutions were performed using deionized water with resistivity of 18.2 MΩ. The catalyst powders (RS−CuNP) were first dissolved in 70% $HNO_3$ (Caledon) and stirred vigorously for 15 min to ensure homogeneous dissolution. The solution was then diluted to reach a $HNO_3$ concentration of 5 wt% using deionized water. To determine the ratio of free-thiols in the catalyst samples, the RS−CuNP powders were washed similar one time with DMF and two times with methanol and centrifuged for 10 min at $5000 \times g$. Both washed and unwashed RS−CuNP powders were then taken through the $HNO_3$ dissolution steps outlined above.

We calculated the percentage of surface atoms in a 25 nm spherical copper nanoparticle as following:

$$V = \frac{4}{3}\pi r^3 = 8.1 \times 10^{-18} \text{cm}^3$$

$$d_{Cu} = 8.96 \text{g/cm}^3$$

$$m_{Cu} = V \times d_{Cu} = 7.3 \times 10^{-17} \text{g}$$

$$MW_{Cu} = 63.55 \text{g/mol}$$

$$n_{Cu, bulk} = 1.15 \times 10^{-18} \text{mol} \times 6.022 \times 10^{23} \text{atom}/mol = 6.9 \times 10^5 \text{atoms}$$

$$A = 4\pi r^2 = 2 \times 10^{-11} \text{cm}^2$$

$$r_{Cu} = 1.28 \text{Å}$$

$$n_{Cu, surface} = \frac{2 \times 10^{-11} \text{cm}^2}{5.1 \times 10^{-16} \text{cm}^2/atom} = 3.9 \times 10^4 \text{atoms}$$

$$\frac{n_{Cu, surface}}{n_{Cu, bulk}} = 5.7\%$$

where $V$, $m_{Cu}$, and $n$ are volume, mass of Cu, and number of atoms, respectively. Based on the percentage of surface atoms, we calculated the coverage (S to surface copper ratio) and plotted in Supplementary Fig. 22.

## Electrochemical characterization

Cyclic voltammetry characterization with electrodes assembled in a flow cell was performed with the same equipment used for CORR experiments. iR correction of the applied potential was not performed. After maintaining −2 V vs. Ag/AgCl for 2 min to ensure the reduction of all the copper oxides, cyclic voltammograms in the non-faradaic region (−0.1 to −0.2 V vs. Ag/AgCl) are performed at 20, 40, 60, 80, 100, and 200 mV/s. ECSA is determined via:

$$C_{dl} = \frac{\frac{1}{2}\left(j_a - j_c\right)}{v} \tag{7}$$

$$ECSA = \frac{C_{dl}}{C_s} \tag{8}$$

where $C_{dl}$ is the double layer capacitance (μF), $j_a$ and $j_c$ are the anodic and cathodic currents, respectively (mA), $v$ is the scan rate (mV/s), and $C_s$ is the specific capacitance of a flat surface with 1 cm² of real surface area (μF/cm²). We assumed the specific capacitance of Cu as $C_s = 29 \, \mu\text{F/cm}^2$ [41].

## In-situ Raman measurements

In-situ Raman was operated with a water immersion objective using a Renishaw inVia Raman microscope. The spectra were collected using a 785 nm laser. To avoid damaging the samples, the full spectrum was collected in two ranges (centered at 700 and 1700 cm⁻¹) using 0.05% full laser intensity, 0.1 s integration time, and 200 repetitions. The raw data were base corrected manually by using Origin 2019 software. An open-structured flow cell was utilized for the measurements. An Ag/AgCl electrode (filled with saturated aqueous KCl solution) and a platinum wire were used as the reference and counter electrode, respectively.

## Computational methods

Our DFT calculations were performed with the Vienna ab initio Simulation Package code[42] using the revised Perdew–Burke–Ernzerhof functional to describe the electron exchange and correlation energy[43]. The interactions between electrons and ion cores were described using the projector augmented wave method[44], while DFT-D3 method was applied to correct the van der Waals interactions[45]. The plane-wave cutoff energy was set to 530 eV. The electronic energy and force convergences were set to $10^{-7}$ eV and 0.02 eV/Å, respectively. The Automated Relaxed Potential Energy Surface Scans[46] method implemented in the Atomic Simulation Environment[47] was used to find the transition states. For catalyst models, we constructed (2 × 2) supercell (111) Cu, which contains two metal atoms in $x$ direction, two metal atoms in $y$ direction and four layers in $z$ direction. Periodic boundary conditions were used in all directions and 18 Å of vacuum layer was used in the $z$ direction to separate the slabs. Monkhorst-Pack 4 × 4 × 1 was utilized to sample the Brillouin zone for the structures. The surface coverage of thiol was set to be 25% in the model of modified Cu catalyst, based on the characterization results from experiments. The two uppermost slab layers and the adsorbates are allowed to relax. The free energy difference of each elementary step was calculated based on the equations as follows:

$$\Delta G = \Delta E + \Delta E_{ZPE} - T\Delta S \tag{9}$$

wherein $\Delta E$ represents total electric energy change; $\Delta E_{ZPE}$ is the change of zero-point energy; $T$ is the temperature (298.15 K); and $\Delta S$ is the difference in entropy. Vibrational frequencies of the adsorbates were calculated using a partial Hessian computed with finite differences, where the adsorbates and thiol molecules were considered, to obtain thermal corrections to the free energy.

## Bader charge analysis

The electronic charge of Cu sites was investigated by Bader charge analysis. Charge on Cu atom is calculated by the difference between the valence electron number ($N_{Valence}$) of Cu and the calculated Bader charges of Cu, as follows:

$$N_{Charge} = N_{Valence} - N_{Bader} \tag{10}$$

Deficiency of electron on surface Cu atoms in thiolated Cu are characterized by positive charges varying from +0.05 e to +0.11 e, while negative charge of −0.02 e on pristine Cu was observed. The Bader charge analysis well explains the observed stronger CO bond to thiolated Cu due to the formation of electron deficient copper upon thiolation.

## Data availability

The data generated in this study are provided in the Supplementary Information and Source Data file. Source data are provided with this paper.

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

## Acknowledgements

This work was supported by the Swiss National Science Foundation (SNSF P2ELP2_199812, E.S. and P500PN_214309, E.S.) and Natural Sciences and Engineering Research Council of Canada (NSERC). This research used resources of the Advanced Photon Source (APS), an Office of Science User Facility operated for the U.S. Department of Energy (DOE) Office of Science by Argonne National Laboratory and was supported by the U.S. DOE under Contract No. DE-AC02-06CH11357, E.H.S and the Canadian Light Source (CLS) and its funding partners. Q.J., and T.J.G. acknowledge the support from the University of Calgary's Canada First Research Excellence Fund Program, the Global Research Initiative in Sustainable Low Carbon Unconventional Resources. S.S. acknowledges the support from Natural Sciences and Engineering Research Council (NSERC) of Canada (Discovery Grant No. RGPIN-2023-05298). The computational research was enabled in part by support provided by computational resource at the University of Calgary (www.rcs.ucalgary.ca) and Compute Canada (www.computecanada.ca). The authors thank D.M.M. and Dr. Zou Finfrock from 20BM beamline at APS, and Dr. Mohsen Shakouri, Dr. Qunfeng Xiao, and Dr. Alisa Paterson from SXRMB at CLS and Dr. Tom Regier and Dr. James Dynes from SGM at CLS for assistance in collecting the X-ray absorption spectroscopy data.

## Author contributions

E.H.S, S.S. and D.S. oversaw the project. E.S. conceptualized the idea, conducted the majority of experiments, and collaborated on writing the article. Q.J. conducted primary DFT calculations and contributed to the article writing. A.S.Z. and R.D. aided in writing and conducted certain experiments on efficiency and stability. T.J.G. contributed to DFT simulations of Raman spectra. P.O. provided assistance with DFT calculations. J.A. and D.M.M. supported XAS experiments. B.H.L., J.Z., J.W., W.N., G.L., C.T. and S.P. assisted in analyzing and performing experiments to gather data on efficiency and NMR. A.S.R. supported XPS characterization. V.B. conducted and analyzed TEM and EELS experiments. All authors participated in discussions regarding the results and provided feedback on the manuscript.

## Competing interests

The authors declare no competing interests.
