## [Peer Review File · Nature Communications]

REVIEWER COMMENTS

Reviewer #1 (Remarks to the Author):

The manuscript by Shirzadi et al. deals with the modification of Cu nanoparticles with alkylthiols for the electroreduction of CO to acetate. The authors report that the lone pairs of thiol groups interact with *CO and promote the hydrogenation of *CO to *CHO. DFT studies also revealed that functionalization selectively stabilizes the intermediate (HOOC-CH₂)*, leading to preferential acetate formation. The functionalization of the Cu surface is first characterized using electron microscopy, XPS and XAS spectroscopies. Raman spectroscopy was then studied to elucidate the interactions between CO adsorbed on the Cu surface of pristine and functionalized nanoparticles. The experimental shift of the Cu-CO vibrational peak from predictions is attributed to weaker C-O and stronger Cu-C bonds in the CO near the thiol.

Overall, the data are compelling and represent another promising example of Cu surface modification for CO/CO₂ electroreduction. I have the following comments for the authors:

1/ Many figures in the supplementary materials are missing, which does not allow for a complete evaluation of the work.

2/ The authors claimed that the alkyl chains are vertically oriented on the Cu surface. This is rather surprising considering the flexibility of alkyl chains. Is it possible to have unbound thiols on the Cu surface?

3/ How did the authors determine the degree of functionalization of the Cu nanoparticles? Based on the ECSA reduction, and given the thickness of the organic layer, I would expect limited access to the Cu surface. Did the authors observe any diffusion limitations?

4/ Some of the examples can be found in the literature on thiol functionalization of Cu (See for example: Ref 9 and Nature Communications 12 (7210), 1-11) with significantly lower activity. The main difference comes from the use of CO instead of CO₂. It would be interesting to compare the catalytic properties in CO₂ and in CO.

5/ It is not clear which potential is uncorrected in the manuscript. It is also unclear how the 5.5% sodium acetate solution was obtained.

Reviewer #2 (Remarks to the Author):

In the work entitled "Nucleophilic surface ligand interaction enables selective CO electroreduction to acetate", Cu particles bound by thiol surface ligands were used as catalysts to reveal the nucleophilic interaction between available lone pairs of sulfur and empty orbitals of reaction intermediates to

promote electrochemical CO to acetate. However, I have many questions about the material characterizations and data interpretations after going through this manuscript and the reviewer would not recommend this article for publication in Nature Communications. In addition, Cu catalysts bound by thiol surface ligands have been used for CORR to acetate in e.g., 2017 (Gong et al., ACS Cent. Sci. 2017, 3, 1032–1040). The specific comments are as follows:

1. This manuscript suggests the nucleophilic interaction between available lone pairs of sulfur in thiol surface ligands and empty orbitals of reaction intermediates is the main reason for the promoted transformation from CO to acetate on Cu. However, the bonding configuration of surface Cu seems different on CuNPs and C18S-CuNPs. One is mainly Cu(II) and the other is mainly Cu(I). Therefore what is the key factor for the different CORR performance of the samples? The valence of the surface Cu? The presence of surface sulfur? Or even the alkane group of the thiol? More control experiments and theoretical discussion should be provided.
2. The DFT calculations in this manuscript are applied on Cu surface. But the surface Cu on CuNPs seems to be Cu(II) before CORR test in this manuscript, while the surface Cu on C18S-CuNPs is Cu(I). Maybe the calculation model can reflect the real reaction mechanism of the samples.
3. The HAADF image in Fig. 2a is in poor quality and also reveals that the thickness of thiols is hardly controlled. One can see the coating is quite nonuniform and in some region it seems to be ~20-30 nm thick, instead of “monolayer assembly” as the author claimed. Such an inhomogeneous thiols coating would make it extremely hard to discuss the effect of adsorbed layers on the CORR performance
4. In this article, the author maintained that the lowest HER value of RS-Cu NPs was due to their hydrophobicity, and studied some catalysts with hydrogen bond receptor/donor functional groups as controls. However, different functional groups can bind to the copper surface with different strengths, thus affecting the ligands surface-coverage. When the surface-coverage of ligands is inconsistent, it is inaccurate to verify the effect of hydrophilicity or hydrophobicity of functional groups on HER value. The surface coverage of the thiols on Cu with different loading should be quantified.
5. In-situ Raman spectra for the samples just provide ambiguous evidence for the possible intermediates. Is there any evidence that can prove the conversion of (HOC-CH₂O)* intermediate to (HOOC-CH₂)*?
6. The authors think EXAFS “proves the contribution of coordination of sulfur to copper in the scattering of ejected photoelectrons”. However, EXAFS mainly reflects the bulk bonding configuration of the sample. Why does the surface Cu demonstrate so obvious contribution for C18S-CuNPs? Moreover, the surface Cu of CuNPs seem to be oxidized to Cu(II). Why doesn't it display any shift?
7. The author selected too limited control samples to study the influence of alkanethiol length, only C10SH is not enough.
8. The sentence in page 8 “an effect that saturates, presumably after a near-complete monolayer is achieved” should be supported by comparing the elemental composition of the spent C18S-Cu NP samples with a load of 5 mg and 15 mg. In addition, the multiple loadings appearing in Fig. 4b are the initial loadings including physisorbed and chemisorbed alkanethiols, which should be mentioned. Because some physisorbed alkanethiols are removed by reacting with alkaline electrolyte during the electrochemical reduction process.
9. In the process of applying self-assembled monolayers to Cu catalysts for CORR, whether the activity for CO reduction will be affected by the change of energy barrier for polar CO molecules diffusion into

the densely packed non-polar self-assembled monolayers?

10. Some mistakes should be revised, e.g., on line 130, "asses" is incorrect; In Fig. 2, caption needs to be corrected ("f, R-space EXAFS spectra of catalysts C18S-Cu NPs and Cu NPs" shall be removed).

11. Way more details about the electrochemical measurements should be given and discussed. How does ECSA be measured? What is the role of Supplementary Figure 22 c-d? In principle, the quasi-linear relationship between scan rate the current density suggests a surface adsorption control process, any discussion or interpretation here? What is the reducing potential in Supplementary Figure 18 and 20? Considering these problems, I think the publication of this manuscript is premature.

Reviewer #3 (Remarks to the Author):

This work mainly put forward the idea of using the lone pairs of sulfur in thiol ligands to regulate the electrochemical CO to acetate on Cu. The authors proved that the alkanethiol adsorbed on the surface of Cu catalyst did play an important role in CORR by using a lot of experiment analysis and theoretical calculation. This manuscript is well-organized, and the experiments and results are reasonable. I would like to ask the authors to major revise the following points.

1. The authors introduce the reduction of CO to CHO* (page 3), and then go directly to the conversion of (HOC-CH₂O)* intermediate to (HOOC-CH₂)* (page 4), where a connection, such as C-C coupling, is missing and suggested to be added.
2. Generally, the catalytic activity of electrocatalysts is related to the number of active sites exposed on the catalyst surface, which can be reflected in the value of ECSAs. The authors mention that "thiol samples had 5x lower effective area than controls", since the ECSA values become smaller, won't it affect the final catalytic activity?
3. The authors observed by STEM images that the surface of the sample was covered with a monolayer of thiols. Since it can be seen on STEM images, it means that the organic ligands on the surface reached a certain concentration, and such a high concentration of coverage may be the reason for the small ECSA value. In addition, the organic ligand contains 18 carbon atoms, wouldn't the high concentration of long alkyl chain coverage hinder the adsorption of reactants on the catalyst surface and thus have an effect on the activity of catalytic CO to produce acetate?
4. The authors concluded that the alkanethiol length has little effect on the performance, so why not use ligands with short carbon chains? Moreover, considering that the authors used C₂SH in the theoretical calculation, it is suggested that the experimental data of C₂SH should be added when comparing the effect of carbon chain length on catalytic performance, the results will be more convincing.
5. Make sure all abbreviations are written out in full the first time used. For example, C18S-CuNPs.
6. When the electrode potentials were converted to the RHE, the authors used a standard potential of 0.235 V for Ag/AgCl. However, in the authors' previous work, 0.197 V was used (ACS Energy Lett. 2021, 6, 79-84), and 0.197 V was used in most of the previously reported works. It is suggested that the authors give a reason for choosing 0.235V.
7. Can alkanethiol be replaced by other ligands containing lone pairs? For example, aliphatic amine.

8. It is recommended to add a performance comparison table.
9. Mismatch between captions and images in Figure 2.
10. There should be a discussion of the work at the end of the article.

We are most grateful for the insightful and valuable comments and questions from the reviewers. By considering these questions, our manuscript is improved, and our detailed answers are given below. The additions and changes to the manuscript are highlighted in yellow in the submitted revised manuscript and supplementary information.

Reviewer1

The manuscript by Shirzadi et al. deals with the modification of Cu nanoparticles with alkylthiols for the electroreduction of CO to acetate. The authors report that the lone pairs of thiol groups interact with *CO and promote the hydrogenation of *CO to *CHO. DFT studies also revealed that functionalization selectively stabilizes the intermediate (HOOC-CH₂)*, leading to preferential acetate formation. The functionalization of the Cu surface is first characterized using electron microscopy, XPS and XAS spectroscopies. Raman spectroscopy was then studied to elucidate the interactions between CO adsorbed on the Cu surface of pristine and functionalized nanoparticles. The experimental shift of the Cu-CO vibrational peak from predictions is attributed to weaker C-O and stronger Cu-C bonds in the CO near the thiol. Overall, the data are compelling and represent another promising example of Cu surface modification for CO/CO₂ electroreduction.

We thank Reviewer's appreciation of our work and for the comments that help us to further improve the manuscript.

Q1: Many figures in the supplementary materials are missing, which does not allow for a complete evaluation of the work.

We are sorry for the missing figures. This is likely due to the conversion from word to pdf during submission process. To avoid this and present you with all figures, we also submitted the pdf version of the manuscript and the supplementary information.

Q2: The authors claimed that the alkyl chains are vertically oriented on the Cu surface. This is rather surprising considering the flexibility of alkyl chains. Is it possible to have unbound thiols on the Cu surface?

We thank the reviewer for this comment. We agree with the reviewer that the alkyl chain would not orient vertically due to its flexibility. However, they would still orient according to their sp^3 hybridization. Regarding the unbound thiols, they are present even after washing steps, as evident from our TEM images. However, we posit that these unbound species will dissolve in KOH - as thiol groups are weak acids - and will form potassium thiolate salts. Supplementary Figure 9 shows the sulfur XPS of spent and fresh samples, and it can be seen that the unbound portion (binding energy >163 eV) of the thiols is no longer present after the reaction.

To explain this point, the following sentences are added on page 5:

“We don’t expect long-chain alkane groups to orient vertically such that even at the thinnest point, unbound alkanethiols are absorbed via hydrophobic interaction prior to CORR. Moreover, (physisorbed) alkanethiols are expected to be removed by the alkaline electrolyte during CORR (Supplementary Figure 6). This happens due to the acidic nature of alkanethiols reacting with KOH electrolyte.”

Q3: How did the authors determine the degree of functionalization of the Cu nanoparticles? Based on the ECSA reduction, and given the thickness of the organic layer, I would expect limited access to the Cu surface. Did the authors observe any diffusion limitations?

We thank the reviewer for this valuable comment. To better understand the degree of functionalization, we prepared additional films with varying loadings. We subsequently measured their contact angle, performance, and electrochemically active surface area (ECSA). Additional experiments have been performed on the different C₁₈SH loadings. These results are summarized in Table R1.

Moreover, based on the XPS spectra, the S-to-Cu ratio is estimated to be ~1:4, suggesting a coverage of 25%. However, the actual coverage is much smaller than this, as the penetration depth of the X-ray beam could reach several Cu layers.

Based on ECSA results, as the C₁₈SH loading increased, the ECSA is reduced, indicating higher degrees of functionalization of the CuNPs. The ECSA reduction is continued until 5 mg loading, and the ECSA did not increase significantly when going from 5mg to 15 mg loading. This indicates the saturation effect that is discussed in the paper. We added extra explanations to page 11, and the loading, ECSA and contact angle results were added to Supplementary Table 7. We did not observe a spike in HER, an indication of diffusion limitation, up to current densities as high as 400 mA/cm² for C₁₈S-CuNPs. In contrast, we expect the alkane-thiols to dissolve CO and thus help the diffusion. The lack of diffusion limitation is also partly attributed to the removal of physisorbed thiols.

Table R1. The ECSA, contact angle, and operating voltage of different loading of C₁₈S-CuNPs.

Loading	Receding Contact angle (degrees)	Advancing Contact angle (degrees)	ECSA (cm ² _{ECSA})	Voltage vs. Ag/AgCl at 100 mA/cm ² (not IR corrected)
Cu NPs (0 mg)	hydrophile	hydrophile	62.0	-1.80
1 mg	162.1	157.5	37.6	-1.70
2 mg	166.9	158.0	23.8	-1.68
5 mg	168.8	163.2	16.7	-1.67
15 mg	169	163	15.5	-1.67

We agree that we cannot calculate the absolute value for the coverage, but we can estimate it relative to each other by looking at ECSA values (Figure R4) : surface coverage = $\frac{ECSA_{CuNP} - ECSA}{ECSA_{CuNP}}$. Please Note the

relatively linear dependence up to 2 mg and saturation effect for 5 and 15 mg of loading; a trend which is also observed in the CORR performance (Figure 4b of the manuscript).

Figure R1. The dependence between $C_{18}SH$ loading and surface coverage based on ECSA.

We revised the manuscript by adding the following sentences on pages 111 and 12. Additionally, the ECSA graphs are added to Supplementary Figures 21-24, and the results are added to Supplementary Table 7 and Supplementary Figure 19.

“Higher loadings of $C_{18}SH$ render the surface more hydrophobic and reduce the ECSA, demonstrative of a high coverage of alkane-thiols (Supplementary Table 7). The ECSA reduction continues until a loading of 5 mg, with no significant change in ECSA at a higher loading of 15 mg (Supplementary Figure 19).”

“Although the surface of copper nanoparticles in RS-CuNPs is well-covered by thiols, in the case of $C_{18}S$ -CuNPs and $C_{12}S$ -CuNPs, the removal of physisorbed thiols is enough to make enough room for the diffusion of reactants.”

Q4: Some of the examples can be found in the literature on thiol functionalization of Cu (See for example: Ref 9 and Nature Communications 12 (7210), 1-11) with significantly lower activity. The main

difference comes from the use of CO instead of CO₂. It would be interesting to compare the catalytic properties in CO₂ and in CO.

We thank the reviewer for the query and suggestion. We have performed CO₂RR with the C₁₈S-CuNPs catalyst and obtained the following results (Table R2). It is worth mentioning that we added Nature Communications 12 (7210), 1-11 to the draft for a better literature review.

Our results indicate the importance of functional groups and the thiol's chemical structure, as they significantly affect the performance under CORR. Moreover, we find the substrate to be of significant importance as well, similar to the report of Nature Communications 12 (7210), 1-11, where a Cu-Ag catalyst was used.

Table R2. Electrochemical performance table of C₁₈S-CuNPs at various current densities in CO₂RR

Current density (mA/cm ²)	H ₂ FE	Ethylene FE	CO FE	Methane FE	Ethanol FE	Propanol FE	Acetate FE	Formate FE
100	7.8±1.7	9.2±0.8	41.1±2.8	0.0±0.0	20.0±1.5	0.0±0.2	3.5±1.1	11.2±1.9
200	7.9±1.4	19.0±1.5	34.9±2.7	0.1±0.0	15.4±1.5	7.4±0.6	2.4±1.4	8.0±1.7
300	7.5±2.0	32.6±2.6	21.9±1.6	0.4±0.2	21.5±2.3	4.5±0.9	4.9±2.2	5.6±1.4
400	22.4±2.4	26.5±2.9	8.7±0.5	2.7±1.0	27.2±1.7	5.0±0.4	8.1±1.6	4.7±1.2

The results of the CO₂RR tests (C₁₈SH) show a significantly lower HER activity than the catalyst reported in Nature Communications 12 (7210), 1-11 that uses C₃SH.

We added the CO₂RR results as Supplementary Table 9, and a paragraph is added to page 12:

“The alkanethiol-covered copper nanoparticles can also be used in CO₂RR as an effective way to obtain high C₂₊ products. Covering Cu NPs with octadecanethiols allows the copper nanoparticles to operate without needing Nafion binders and have current densities as high as 300 mA/cm² (Supplementary Table 9).

Q5: It is not clear which potential is uncorrected in the manuscript. It is also unclear how the 5.5% sodium acetate solution was obtained.

We thank the reviewer for the comment. We have added the corrected/ uncorrected potentials to the revised manuscript. We have also modified the experimental section to explain how the 5.5% sodium acetate was obtained in the stability test. This high concentration was obtained by performing CORR in an MEA equipped with a CEM to decrease the liquid products crossover to the anode chamber. The liquid products in the cathode side were collected with a liquid trap (2-6°C), resulting in a sodium acetate concentration as high as 5.5 weight percent.

Reviewer2

In the work entitled “Nucleophilic surface ligand interaction enables selective CO electroreduction to acetate”, Cu particles bound by thiol surface ligands were used as catalysts to reveal the nucleophilic interaction between available lone pairs of sulfur and empty orbitals of reaction intermediates to promote electrochemical CO to acetate. However, I have many questions about the material characterizations and data interpretations after going through this manuscript and the reviewer would not recommend this article for publication in Nature Communications. In addition, Cu catalysts bound by thiol surface ligands have been used for CORR to acetate in e.g., 2017 (Gong et al., ACS Cent. Sci. 2017, 3, 1032–1040).

We appreciate the reviewers' feedback, which enabled us to further improve the manuscript. To the best of our knowledge, our work is the first in developing thiols-Cu nanoparticles catalyst in CORR for selective production of acetate at high current densities. Gong et al. (ACS Cent. Sci. 2017, 3, 1032–1040) developed a hetero-bimetallic molecular materials by assembly of the C₂-linked Fe porphyrin derivative on Cu. Their catalyst could obtain up to 57% ethanol and 24% acetate at –0.4 V vs RHE and a current density of 1.34 mA/cm². They show that the Fe center in the molecular additive can aid in cooperative reduction of potential acetaldehyde intermediates. Although this work demonstrated a significant improvement in the C₂ product generation compared to the unmodified copper, this was achieved at low current densities (1.34 mA/cm²) in H-cell and cannot be extrapolated to industry-relevant current densities. In our work, we have studied the nucleophilic interaction between available lone pairs of sulfur in thiol surface ligands, and empty orbitals of reaction intermediates which selectively promote electrochemical CO to acetate on copper. Our catalyst design resulted in an acetate FE of 70% at 400 mA/cm² and C₂₊ FE of 98% thus, it can be concluded that Cong et al. work, although significant work, has a different scope than ours.

Q1: This manuscript suggests the nucleophilic interaction between available lone pairs of sulfur in thiol surface ligands and empty orbitals of reaction intermediates is the main reason for the promoted transformation from CO to acetate on Cu. However, the bonding configuration of surface Cu seems different on CuNPs and C₁₈S-CuNPs. One is mainly Cu(II) and the other is mainly Cu(I). Therefore what is the key factor for the different CORR performance of the samples? The valence of the surface Cu? The presence of surface sulfur? Or even the alkane group of the thiol? More control experiments and theoretical discussion should be provided.

We thank the reviewer's comment, which motivates us to further study the difference between Cu and different thiol-modified coppers in CORR for acetate production. Although before the reaction, the CuNP is quite oxidized to Cu(II), and the surface of C₁₈S-CuNPs is mainly Cu(I). However, under reduction, we expect all the Cu(II) and Cu(I) to convert to metallic copper **for CuNPs**. In the case of **C₁₈S-CuNPs**, only the copper atoms attached to thiols maintain their Cu(I) oxidation state. This way it maintains the charge neutrality condition (thiolates have a negative charge and this means the attached copper should have +1 oxidation state assuming only one thiol is attached to the copper atoms). However, the oxidation state of copper atoms that attach to CO_(adsorbed) is 0. Therefore, under reduction conditions, copper atoms in C₁₈S-CuNPs have an oxidation state of 0 except the ones attached to thiol surface ligands. We added this to page 7.

“It is important to mention that prior to the reaction, the surface of CuNPs is predominantly made up of Cu^{II} and the surface of C₁₈S-CuNPs are mainly Cu^I. However, under the reductive potential of the reaction condition we expect the surface of the CuNPs to reduce to Cu⁰. In the case of C₁₈S-CuNPs, only the copper atoms attached to the thiols maintain their Cu(I) oxidation state, while the non-bound Cu atoms are reduced to Cu⁰, allowing for charge neutrality to be held at all times.”

We have performed more control experiments to specifically study the effect of alkane chain length on CORR performance (See Table R3). C₅S-Cu NPs, C₁₂S-Cu NPs, C₁₈S-Cu NPs and C₂₂S-Cu NPs showed a similar acetate FE, indicating alkyl chain length to not play a role in the high acetate FE in modified copper catalyst. Moreover all the R_s-Cu NPs showed higher acetate FEs than CuNPs, demonstrating the promotional effect of thiols for acetate production.

Table R3. Electrochemical performance table of different alkanethiols at 100 mA/cm²

Catalyst	H ₂ FE (%)	C ₂ H ₄ FE (%)	CH ₄ FE (%)	C ₂ H ₅ OH FE (%)	C ₃ H ₇ OH FE (%)	CH ₃ COO ⁻ FE (%)
C ₂ S-Cu NPs	27.3±4.2	22.9±0.9	0.0±0.0	16.9±2.0	3.0±0.7	24.1±2.2
C ₅ S-Cu NPs	7.1±1.2	24.3±1.3	0.9±0.2	10.2±1.4	3.3±1.3	40.8±3.6
C ₁₂ S-Cu NPs	7.6±1.6	18.7±2.0	0.0±0.0	12.6±1.7	12.2±1.9	45.4±3.7
C ₁₈ S-Cu NPs	5.6±1.5	19.2±2.5	0.0±0.0	13.4±1.3	8.2±0.9	46.1±3.2
C ₂₂ S-Cu NPs	5.0±1.2	26.5±2.8	0.0±0.0	13.0±2.2	4.3±1.1	37.7±3.0
Cu NPs	39.7±2.3	10.8±1.2	0.0±0.0	7.4±1.8	5.4±0.8	13.5±1.9
Cu NPs+Nafion	9.4±1.1	37.8±2.3	0.0±0.0	13.6±1.5	15.3±2.1	14.4±1.3

As a result of the reviewer’s suggestions, we performed more control experiments. The alkane chain length significantly affected the performance of CORR as it controls the diffusion. A few theoretical control experiments are provided to support the effective interaction of lone pairs with CORR intermediates required to stabilize them. We replace the -CH₂CH₂CH₃ instead of -SCH₂CH₃ in our theoretical experiments (Figure R1). This way, no lone pair is available to interact with the intermediates. Surface CH₃CH₂CH₂- did not stabilize CHO*; in contrast, it destabilized the CHO* by 1.09 eV. We also used CH₃CH₂O- and CH₃CH₂Se- as they are in the same group as sulfur and have available lone pairs. As shown in Figure R2, oxygen destabilizes the CHO* intermediate by 1.02 eV. This is because oxygen lone pairs did not have any interaction/bond with CORR intermediates (the distance between O of ethanolate ligand and C of CO* and CHO* is higher than their combined atomic radius). This suggests that the copper atom that the ligand is attached to and has the oxidation state of +1 does not impact the stabilization of the CORR intermediates and effective interacting lone pairs are essential for this purpose. Selenium in CH₃CH₂Se-ligand is able to interact with CORR intermediates (Figure R3). However, the interaction is not as effective and strong as sulfur. This interaction resulted in a slight stabilization of 0.04 eV compared to a clean slab (0.36 eV vs. 0.4 eV for CH₃CH₂Se-covered and clean slabs). **This indicates sulfur has the most optimum interaction due to its size and electronics and effective lone pair interaction is the key to high CORR performance.** All these control experiments had CH₃CH₂ in common, and therefore alkane group could not be the reason for the high performance of R_S-CuNPs.

We have also performed the Cu modification with oleylamine as a ligand, however there is no chemical attachment of the ligand to the copper under our synthesis and experimental conditions.

The following sentences have been added to pages 4 and 5, and the new DFT calculations were added to the supplementary information.

“To illustrate the significance of the sulfur interaction, the $-\text{SCH}_2\text{CH}_3$ ligand is replaced with $-\text{CH}_2\text{CH}_2\text{CH}_3$ (Supplementary Figure 2). The $-\text{CH}_2\text{CH}_2\text{CH}_3$ ligand did not stabilize the CORR RDS, and in fact, destabilized the formation of CHO^* by 1.09 eV. We also investigated other elements in the periodic table with lone pairs similar to that of sulfur. As shown in Supplementary Figure 3, oxygen destabilizes the CHO^* intermediate by 1.02 eV and does not interact with other CORR intermediates (the distance between O of ethanolate ligand and C of CO^* and CHO^* is higher than their combined atomic radius). This suggests that the copper atom attached to the ligand with an oxidation state of +1 has no impact on the stabilization of the CORR intermediates, and effective interacting lone pairs are essential for this purpose. Selenium in the $\text{CH}_3\text{CH}_2\text{Se}$ -ligand is able to interact with CORR intermediates (Supplementary Figure 4). However, the interaction is less effective and strong than sulfur. This interaction resulted in a slight stabilization of 0.04 eV compared to a clean slab (0.36 eV vs. 0.4 eV for $\text{CH}_3\text{CH}_2\text{Se}$ -covered and clean slabs). These results show the importance of atomic radius and lone pairs in sulfur’s ability to interact with CORR intermediates and stabilize them.”

Figure R1. The hydrogenation of CO* adsorbate to CHO* over copper with propyl carbanion attached.

Figure R2. The hydrogenation of CO* adsorbate to CHO* over copper with ethanolate attached.

Figure R3. The hydrogenation of CO* adsorbate to CHO* over copper with ethyl selenolate attached.

Q2: The DFT calculations in this manuscript are applied on the Cu surface. But the surface Cu on CuNPs seems to be Cu(II) before CORR test in this manuscript, while the surface Cu on C18S-CuNPs is Cu(I). Maybe the calculation model can reflect the real reaction mechanism of the samples.

The thin layer of oxide in the copper nanoparticles is being reduced by applying a reducing potential. Therefore, we expect all the copper oxides convert to copper (0) except the ones attached to the sulfur atom of thiols which remain as Cu(I). The calculations have already considered the oxidation state of ligated copper atoms.

Q3: The HAADF image in Fig. 2a is in poor quality and also reveals that the thickness of thiols is hardly controlled. One can see the coating is quite nonuniform and in some region it seems to be ~20-30 nm thick, instead of “monolayer assembly” as the author claimed. Such an inhomogeneous thiols coating would make it extremely hard to discuss the effect of adsorbed layers on the CORR performance.

Thanks for pointing this out. We have replaced all the poor-quality images with high dpi images. We have also added a higher magnification STEM image of C₁₈S-CuNPs to the supplementary information (supplementary Figures 6c and d). There is an accumulation of unbound (physisorbed) thiols in some regions of CuNP due to the hydrophobic interaction of alkyl chains. These physisorbed thiols are expected to be removed in the presence of a circulating electrolyte (KOH) during the CORR (RSH is a weak acid and reacts with KOH and therefore is washed away).

We have also performed CORR using CuNP with different loadings of C₁₈SH. A higher loading would increase the unbound (physisorbed) thiols. We have observed that high loadings (>5mgC₁₈SH/10mg CuNP) have little to no effect on the performance, confirming the role of chemisorbed thiols as the active species.

Even with a loading of 15 mg, we could reach current densities up to 400 mA/cm², indicating the physisorbed thiols did not impact the performance (Table R4).

Table R4. Electrochemical performance table of C₁₈S-CuNP with the loading of 15mg/10mg CuNP at various current densities.

Current density (mA/cm ²)	H ₂ FE (%)	C ₂ H ₄ FE (%)	CH ₄ FE (%)	C ₂ H ₅ OH FE (%)	C ₃ H ₇ OH FE (%)	CH ₃ COO ⁻ FE (%)
100	5.6±1.5	19.2±2.5	0.0±0.0	13.4±1.3	8.2±0.9	46.1±3.2
200	6.1±1.3	20.0±1.8	0.0±0.0	10.3±1.1	7±1.0	58.3±4.1
300	5.1±0.8	20.9±2.4	0.0±0.0	6.9±0.7	0.8±0.8	66.2±5.2
400	5.1±0.8	22.1±3.1	0.0±0.0	5.4±0.9	0.9±0.4	69.5±4.8

We have revised the manuscript on pages 5 and 12 and accordingly.

Q4: In this article, the author maintained that the lowest HER value of RS-Cu NPs was due to their hydrophobicity, and studied some catalysts with hydrogen bond receptor/donor functional groups as controls. However, different functional groups can bind to the copper surface with different strengths, thus affecting the ligands surface-coverage. When the surface-coverage of ligands is inconsistent, it is inaccurate to verify the effect of hydrophilicity or hydrophobicity of functional groups on HER value. The surface coverage of the thiols on Cu with different loading should be quantified.

We agree with the reviewer that different ligands would interact differently and could lead to different coverages (see table R3). We observed different coverages throughout C₂₂SH to C₂SH. As a shorter alkyl chain alkyl group is used, a higher coverage is observed based on the hydrophobicity test and ECSA experiments. It seems C₁₈ and C₁₂ had the most optimum coverages. These explanations are added to pages 10 and 11.

We have revised the manuscript by adding the possible impact of ligands surface coverage on the HER. However, higher hydrophobicity can lead to better stabilities.

“As controls we screened other ligand surface modifiers that possesses hydrogen bond acceptor/donor functional groups, with all samples showing a higher HER activity (see Supplementary Table 3). **However, the higher HER can be due to different coverage of thiols in these catalysts.**”

Unfortunately, we couldn't use ECSA in 2-mercapto ethanol-covered films for estimating the coverage as such film exhibited an unusually high ECSA (89.7 cm²). In other words, the microenvironment around 2-mercapto ethanol-covered and alkyl thiol-covered films are extremely different. Thus, we cannot employ assumptions that were used in ECSA to determine the coverage.

Table R5. The ECSA, contact angle, and operating voltage of different loading of C₁₈S-CuNPs.

Loading	Receding Contact	Advancing Contact	ECSA (cm ² _{ECSA})	Voltage vs. Ag/AgCl at 100 mA/cm ² (not IR corrected)

	angle (degrees)	angle (degrees)		
Cu NPs (0mg)	hydrophile	hydrophile	62.0	-1.80
1 mg	162.1	157.5	37.6	-1.70
2 mg	166.9	158.0	23.8	-1.68
5 mg	168.8	163.2	16.7	-1.67
15 mg	169	163	15.5	-1.67

As a higher loading of C₁₈SH is used, the surface becomes more hydrophobic, and the ECSA is reduced, indicating higher coverages (Table R5). The ECSA reduction is continued until 5 mg loading, and the ECSA did not significantly change when using a higher 15 mg of loading. This indicates the saturation effect that is discussed in the paper.

We agree that we cannot calculate the absolute value for the coverage, but we can estimate it relatively across samples by looking at ECSA values (Figure R5): surface coverage = $\frac{ECSA_{CuNP} - ECSA}{ECSA_{CuNP}}$. Please Note the relatively linear dependence up to 2 mg and saturation effect for 5 and 15 mg of loading; a trend which is also observed in the CORR performance (Figure 4b of the manuscript).

We included Figure R5 in the manuscript as Supplementary Figure 19. We revised the manuscript by adding the following sentences to page 11, and the loading ECSA and contact angle results were added to Supplementary Table 7:

“Higher loadings of C₁₈SH render the surface more hydrophobic and reduce the ECSA, demonstrative of a high coverage of alkane-thiols (Supplementary Table 7). The ECSA reduction continues until a loading of 5 mg, with no significant change in ECSA at a higher loading of 15 mg (Supplementary Figure 19).”

Figure R5. The dependence between C₁₈SH loading and surface coverage based on ECSA.

Q5: In-situ Raman spectra for the samples just provide ambiguous evidence for the possible intermediates. Is there any evidence that can prove the conversion of (HOC-CH₂O)* intermediate to (HOOC-CH₂)*?

We thank the reviewer for valuable comment. As the conversion of (HOC-CH₂O)* intermediate to (HOOC-CH₂)* happens after the RDS, their coverages are very low, making it is difficult to detect signals associated with either species. However, we managed to capture the Raman peak related to the double bond of (HOOC-CH₂)* intermediate on CuNPs and C₁₈S-CuNPs via Raman spectroscopy. (HOC-CH₂O)* has no distinctive bond, such as a double bond, to be detected by Raman spectroscopy. We realized the peak related to (HOOC-CH₂)* on copper is red-shifted by almost 200 cm⁻¹, indicating the sulfur lone interaction with the empty orbitals of -COOH. We have updated the manuscript with these new results and added Figure 3c. we revise the manuscript with discussions on pages 7 to 8.

“The peak located at ~1600 cm⁻¹ assigned to the -COO in (HOOC-CH₂)* intermediate at ~1600cm⁻¹ is redshifted to ~1400 cm⁻¹ on C₁₈SH-CuNPs (Figure 3c). DFT calculations point to a lateral interaction between sulfur lone pairs and the -COO in (HOOC-CH₂)*, weakening the C-O bond and shifting its vibrational frequency to lower wavenumbers. “

Figure R6. In-situ Raman spectra of C18S-CuNPs show a redshift for the vibration of HOOC-CH₂, suggesting the interaction of sulfur lone pairs with this intermediate.

Q6: The authors think EXAFS “proves the contribution of coordination of sulfur to copper in the scattering of ejected photoelectrons”. However, EXAFS mainly reflects the bulk bonding configuration of the sample. Why does the surface Cu demonstrate so obvious contribution for C18S-CuNPs? Moreover, the surface Cu of CuNPs seem to be oxidized to Cu(II). Why doesn’t it display any shift?

We agree with the reviewer that XAS is mainly a bulk characterization technique. However, we used a fluorescence detector rather than the absorption detector, which is more sensitive to the surface. Moreover, the contribution of sulfur in the EXAFS is small mainly because lots of the signal are coming from the bulk; despite that, we couldn’t fit the EXAFS with a clean copper slab simulation, and it is necessary to include sulfur in the calculation from the relaxed DFT slabs (see Figures R7 and R8). This argument is added to page 7. We tried two different beamlines, one in the Argonne national lab and one in the Canadian light source (CLS), and in both, we observed the shifts in the EXAFS regions of C₁₈S-CuNP. Therefore, we are confident in our results.

We have peaks assigned to Cu-O in EXAFS in the regions around 1.5 Å. However, the contribution of oxygen doesn’t affect the Cu-Cu signal at ~2.2. Since the atomic radius of sulfur (100 pm) is similar to the atomic radius of Cu (128 pm), the Cu-S contribution is mixed with Cu-Cu signal and resulted in a shift rather than a separate peak evolution. Moreover, we expect a bigger radius for sulfur in our case as it is negatively charged; thus, it approaches the Cu atomic size even more. More elaboration is added to page 7.

We revised the manuscript with the following sentences:

“Although XAS is predominantly a bulk analysis technique, the high coverage of thiols on the surface and the use of the more sensitive fluorescence detector, allowed for the detection of the sulfur signal in the R-space of EXAFS. However small, this signal produced a noticeable shift in the Cu-Cu region at ~ 2.2 Å. Since copper and sulfur have similar atomic radii and the sulfur contribution is small, we observe it as a shift in the Cu-Cu region rather than a separate peak.”

Figure R7. R-space EXAFS spectra of $C_{18}S$ -Cu NPs Fit considering the sulfur contribution.

Figure R8. *R*-space EXAFS spectra of C₁₈S-Cu NPs. No valid fit was obtained without including sulfur.

Q7: The author selected too limited control samples to study the influence of alkanethiol length, only C10SH is not enough.

We appreciate the reviewer for this valuable comment. We included C₂SH, C₅SH, C₁₂SH, and C₂₂SH, and the data is added to the paper. To better understand the impact of alkane chain length on the performance, we prepared additional films with different alkyl group chains with similar molar loadings (3.5 mM). Then, their contact angle, performance, and electrochemically active surface area (ECSA) were measured. These results are summarized in Tables R5-R11 and explained below:

C₁₂S-CuNPs and C₁₈S-CuNPs behaved similarly (Table R6 and R7). The C₅SH-covered catalyst exhibited a huge limitation in the current density as we couldn't reach 200 mA/cm² without having high HER Faradaic efficiencies. We think this is because of the high coverage of smaller thiols and their dense packing. This effect was more severe when we made the films with C₂SH thiols as we had a higher HER compared to the C₅SH-covered catalyst even at 100 mA/cm², although this could be partially due to the inability of short (ethyl) chains to help establish proper triple phase boundaries. For C₂₂SH-covered films, we also observed high HER at 100 mA/cm². However, we don't think this necessarily originates from high coverage but from its large alkyl chains that limit the ions and reactant diffusion. This is evident from the higher operating voltages at the same current density (see table R8).

C₂S-CuNPs to C₂₂S Cu NPs showed a superhydrophobic surface, making it impossible to measure the static contact angle as the water drop moved around. For this reason, we presented advancing and receding contact angles in table R8. The static contact angle lies between these two values. We did not observe any significant trend/ difference in the contact angle of C₅S-CuNPs to C₁₂S Cu NPs samples. However, although C₂S-CuNPs showed a superhydrophobic surface, we were not able to measure the receding contact angle

for this sample as the surface got wet after the force was applied. This can be justified considering the surface of C₂S-CuNPs is only covered by a very short (ethyl) alkane chain which cannot repel water under pressure.

ECSA was measured for all the RS-CuNPs to better understand the coverage. We assumed that since the surface is ligated by alkanethiols, it would block the ions from reaching the surface and therefore reduce the ECSA. However, this assumption is not correct in the case of C₂S-CuNPs as an alkyl chain of only two carbon cannot be as effective in blocking the ions as longer chain ones; this was evident from its higher ECSA compared to other RS-CuNPs, but still, the ECSA value is about the half of naked CuNPs. As shown in Table R8, C₁₂S-CuNPs and C₁₈S-CuNPs have similar ECSA, and C₅S-CuNPs have slightly lower ECSA. We think this is because the smaller alkanethiols can pack more densely. The lowest ECSA belongs to the C₂₂S-CuNPs. This can be due to the extremely long alkane chains that inhibit the diffusion of ions reaching the surface and participating in the ECSA.

Table R5. Electrochemical performance table of different alkanethiols at 100 mA/cm².

Catalyst	H ₂ FE (%)	C ₂ H ₄ FE (%)	CH ₄ FE (%)	C ₂ H ₅ OH FE (%)	C ₃ H ₇ OH FE (%)	CH ₃ COO ⁻ FE (%)
C ₂ S-Cu NPs	27.3±4.2	22.9±0.9	0.0±0.0	16.9±2.0	3.0±0.7	24.1±2.2
C ₅ S-Cu NPs	7.1±1.2	24.3±1.3	0.9±0.2	10.2±1.4	3.3±1.3	40.8±3.6
C ₁₂ S-Cu NPs	7.6±1.6	18.7±2.0	0.0±0.0	12.6±1.7	12.2±1.9	45.4±3.7
C ₁₈ S-Cu NPs	5.6±1.5	19.2±2.5	0.0±0.0	13.4±1.3	8.2±0.9	46.1±3.2
C ₂₂ S-Cu NPs	5.0±1.2	26.5±2.8	0.0±0.0	13.0±2.2	4.3±1.1	37.7±3.0
Cu NPs	39.7±2.3	10.8±1.2	0.0±0.0	7.4±1.8	5.4±0.8	13.5±1.9
Cu NPs+Nafion	9.4±1.1	37.8±2.3	0.0±0.0	13.6±1.5	15.3±2.1	14.4±1.3

Table R6. Electrochemical performance table of C₁₂S-CuNPs at various current densities.

Current density (mA/cm ²)	H ₂ FE	Ethylene FE	Methane FE	Ethanol FE	Propanol FE	Acetate FE
100	7.6±2.1	18.7±2.9	0.0±0.0	12.6±1.4	12.2±2.4	45.4±3.3
200	7.2±1.9	20.1±2.0	0.0±0.0	8.4±1.1	9.1±1.9	50.9±3.7
300	6.6±1.4	22.2±3.1	0.0±0.0	8.8±1.4	3.1±1.0	60.1±4.2
400	5.7±1.3	23.7±2.7	0.0±0.0	3.9±1.0	0.6±0.3	67.9±3.5

Table R7. Electrochemical performance table of C₁₈S-CuNPs at various current densities.

Current density (mA/cm ²)	H ₂ FE (%)	C ₂ H ₄ FE (%)	CH ₄ FE (%)	C ₂ H ₅ OH FE (%)	C ₃ H ₇ OH FE (%)	CH ₃ COO ⁻ FE (%)
100	5.6±1.5	19.2±2.5	0.0±0.0	13.4±1.3	8.2±0.9	46.1±3.2
200	6.1±1.3	20.0±1.8	0.0±0.0	10.3±1.1	7±1.0	58.3±4.1
300	5.1±0.8	20.9±2.4	0.0±0.0	6.9±0.7	0.8±0.8	66.2±5.2
400	5.1±0.8	22.1±3.1	0.0±0.0	5.4±0.9	0.9±0.4	69.5±4.8

Table R8. The ECSA, contact angle, and operating voltage of the RS-CuNPs.

Catalyst	Advancing Contact angle (degrees)	Receding Contact angle (degrees)	ECSA (cm ² _{ECSA})	Voltage vs Ag/AgCl at 100mA/cm ² (not IR corrected)
C ₂ S-Cu NPs	163.4	0	29.3	-1.63
C ₅ S-Cu NPs	170.3	163.7	8.3	-1.64
C ₁₂ S-Cu NPs	166.7	162.8	16.5	-1.66
C ₁₈ S-Cu NPs	169	163	15.5	-1.67
C ₂₂ S-Cu NPs	169.6	161	5.0	-1.74
Cu NPs	hydrophile	hydrophile	62.0	-1.80
Cu NPs+Nafion	-	-	96.6	-1.75

We updated Supplementary Table 3 with new C₂₂ to C₂ thiols and added Supplementary Tables 4-6. Additionally, the ECSA graphs are added as Supplementary Figures 21-24. We revised the manuscript with the following discussion:

“Next, we studied the impact of alkane-thiol chain length (Supplementary Table 3) from ethane (C₂) to docosane (C₂₂). We did not observe spikes in hydrogen evolution reaction (HER) indicative of diffusion limitations up to current densities of 400 mA/cm² for C₁₈S-CuNPs and C₁₂S-CuNPs (See Supplementary Tables 4 and 5). However, the C₅S-CuNPs exhibited a considerable limitation in CORR current density, with HER FE values spiking to > 60% at 200 mA cm⁻². This is attributed to the dense packing capacity of smaller thiols and their subsequent high coverage. This effect is exaggerated on even shorter C₂SH thiols, where HER activity was higher compared to C₅S-CuNPs. This may also be due to the inability of short ethyl chains to establish a proper triple-phase boundary. On higher chain length samples such as C₂₂S-CuNPs, an increase in HER is also observed at 100 mA/cm². Here, we posit that the large alkyl chains limit the ion and reactant diffusion onto the surface, inhibiting CORR activity. This is evident from the higher operating voltages at the same current density (Supplementary Table 6). Nevertheless, all C₂ to C₂₂ alkanethiols showed better FEs for acetate than CuNPs.

The mid chain length samples C₅S-CuNPs to C₁₂S Cu NPs showed a superhydrophobic surface, with little change in contact angles across samples (Supplementary Table 6). Although the C₂S-CuNPs also exhibited a superhydrophobic surface, we were not able to measure the receding contact angle for this sample as the surface got wet after the force was applied. This can be justified considering the surface of C₂S-CuNPs is only covered by a very short alkane chain which cannot repel water under pressure.

To better understand the alkane-thiol coverage, the ECSA was measured for all the RS-CuNPs samples (Supplementary Table 6). We posit there to be a direct relation between the ECSA

and alkanethiol coverage such that the alkanethiolated sites do not contribute to the ECSA measurement. Moreover, the alkane-thiols would block ions from reaching the surface and therefore reduce the ECSA further.

On the C₂S-CuNPs sample, the alkyl chain of only two carbons is not as effective in blocking the ions, evident from its higher ECSA (29.3 cm²_{ecsa}) compared to other RS-CuNPs. The ECSA of C₂S-CuNPs catalyst is almost half of bare CuNPs (62 cm²_{ecsa}). As shown in Supplementary Table 6, C₁₂S-CuNPs and C₁₈S-CuNPs have similar ECSA (16.5 and 15.5 cm²_{ecsa}, respectively) and C₅S-CuNPs has a slightly lower ECSA (8.3 cm²_{ecsa}), attributed to the denser packing of the smaller alkane-thiols (C₅ vs. C₁₂ and C₁₈). The C₂₂S-CuNPs had an ECSA of only 5 cm²_{ecsa}, due to the long alkane chains inhibiting ion diffusion to the surface.”

Q8: The sentence in page 8 “an effect that saturates, presumably after a near-complete monolayer is achieved” should be supported by comparing the elemental composition of the spent C18S-Cu NP samples with a load of 5 mg and 15 mg. In addition, the multiple loadings appearing in Fig. 4b are the initial loadings including physisorbed and chemisorbed alkanethiols, which should be mentioned. Because some physisorbed alkanethiols are removed by reacting with alkaline electrolyte during the electrochemical reduction process.

Many thanks for your comment and suggestion. We added an argument including physisorbed and chemisorbed alkanethiols to page 5. Additionally, we agree that basic electrolyte reacts with the physisorbed thiols.

The sentence on page 8, “an effect that saturates, presumably after a near-complete monolayer is achieved,” is now supported by the contact angle and ECSA measurements (Table R9). The results show that increasing the loading from 5 to 15 mg does not change the contact angle nor ECSA noticeably, which is an indication of near-saturation. These arguments are added to the manuscript on page 11. Moreover, we change the “near-complete” to near-saturated”.

Additionally, as it is shown in Figure R5, ECSA can give a good estimate of the coverage and also saturation loading.

We included Figure R5 in the manuscript as Supplementary Figure 19. We revised the manuscript by adding the following sentences to page 11, and the loading ECSA and contact angle results were added to Supplementary Table 7:

“Higher loadings of C₁₈SH render the surface more hydrophobic and reduce the ECSA, demonstrative of a high coverage of alkane-thiols (Supplementary Table 7). The ECSA reduction continues until a loading of 5 mg, with no significant change in ECSA at a higher loading of 15 mg (Supplementary Figure 19).”

We revised the manuscript by mentioning that the initial loading includes physisorbed and chemisorbed thiols to page 11:

“It is worth mentioning that the initial loading is not the same as the loading during the electrochemical reaction as we expect the physisorbed alkanethiols to be removed in alkaline conditions.”

Table R9. The ECSA, contact angle, and operating voltage of different loading of C₁₈S-CuNPs.

Loading	Receding Contact angle (degrees)	Advancing Contact angle (degrees)	ECSA (cm ² _{ECSA})	Voltage vs. Ag/AgCl at 100 mA/cm ² (not IR corrected)
Cu NPs (0mg)	hydrophile	hydrophile	62.0	-1.80
1 mg	162.1	157.5	37.6	-1.70
2 mg	166.9	158.0	23.8	-1.68
5 mg	168.8	163.2	16.7	-1.67
15 mg	169	163	15.5	-1.67

Q9: In the process of applying self-assembled monolayers to Cu catalysts for CORR, whether the activity for CO reduction will be affected by the change of energy barrier for polar CO molecules diffusion into the densely packed non-polar self-assembled monolayers?

We thank the reviewer for this valuable comment. The CO molecules are relatively apolar (with only a dipole moment of 0.122D, due to the dative bond between O and C that makes the formal charges against the charges expected from their electronegativity), and studies show CO is more soluble in apolar alkanes and organic solvents than water¹. Thus, CO is expected to be well dissolved into alkane chains of alkanethiols. However, CO diffusion depends on the packing of chemisorbed thiols.

The coverage depends on which alkane thiol is used. Based on our ECSA results, C₅S-Cu NPs have one of the highest packings. C₂₂S-CuNPs also have a low ECSA, possibly due to long alkane chains acting as insulators. Without alkane thiols that dissolve CO molecules, we cannot reach current densities higher than 50 mA/cm² without using a binder like Nafion. In fact, alkanethiols facilitate CO availability on the Cu surface for CORR reaction, as is the case for CO₂RR (please see Reviewer 1, comment 4). We added this to page 12.

“CO molecules are weakly polar (with only 0.122D), and hence more soluble in apolar alkanes and organic solvents than water²⁵. Thus, CO is expected to be well dissolved into alkane chains of alkanethiols. However, CO diffusion is also controlled by the packing of chemisorbed thiols. Without alkane-thiols that dissolve CO molecules, we cannot reach current densities higher as high

¹ Jáuregui-Haza, U. J., et al. "Solubility of hydrogen and carbon monoxide in water and some organic solvents." *Latin American applied research* 34.2 (2004): 71-74.

as 100 mA/cm² without using a binder like Nafion (Supplementary Table 3). Alkanethiols facilitate CO availability on the Cu surface for CORR reaction, as was the case for CO₂RR.”

Q10: Some mistakes should be revised, e.g., on line 130, “asses” is incorrect; In Fig. 2, caption needs to be corrected (“f, R-space EXAFS spectra of catalysts C18S-Cu NPs and Cu NPs” shall be removed).

Thanks for the comments. We have revised the manuscript accordingly.

Q11: Way more details about the electrochemical measurements should be given and discussed. How does ECSA be measured? What is the role of Supplementary Figure 22 c-d? In principle, the quasi-linear relationship between scan rate the current density suggests a surface adsorption control process, any discussion or interpretation here? What is the reducing potential in Supplementary Figures 18 and 20?

Thanks for the comment. A detailed methodology on the ECSA has been added to the manuscript. The ECSA discussion has been updated with extra experiments (see methods under electrochemical characterization). More discussion on the ECSA is added to pages 10 and 11. The reducing potential in Supplementary Figures 18 and 20 is -1.7 V vs. Ag/AgCl (not iR corrected) and added to the manuscript.

We revised the manuscript by adding the following to the experimental section:

“After maintaining -2 V vs. Ag/AgCl for 2 mins to ensure the reduction of all the copper oxides, cyclic voltammograms in the non-faradaic region (-0.1- to -0.2 V vs Ag/AgCl) are performed at 20, 40, 60, 80, 100, and 200 mV/s. electrochemically active surface area (ECSA) is determined via:

$$C_{dl} = \frac{\frac{1}{2}(j_a - j_c)}{v},$$
$$ECSA = \frac{C_{dl}}{C_s},$$

where C_{dl} is the double layer capacitance (μF), j_a and j_c are the anodic and cathodic currents, respectively (mA), v is the scan rate (mV/s), and C_s is the specific capacitance of a flat surface with 1 cm² of real surface area ($\mu\text{F}/\text{cm}^2$). We assumed the specific capacitance of Cu as $C_s = 29 \mu\text{F}/\text{cm}^2$.”

Reviewer3

This work mainly put forward the idea of using the lone pairs of sulfur in thiol ligands to regulate the electrochemical CO to acetate on Cu. The authors proved that the alkanethiol adsorbed on the surface of Cu catalyst did play an important role in CORR by using a lot of experiment analysis and theoretical calculation. This manuscript is well-organized, and the experiments and results are reasonable.

We thank the Reviewer’s appreciation of our work and for the comments that help us to further improve the manuscript.

Q1. The authors introduce the reduction of CO to CHO* (page 3), and then go directly to the conversion of (HOC-CH₂O)* intermediate to (HOOC-CH₂)* (page 4), where a connection, such as C-C coupling, is missing and suggested to be added.

Many thanks to the reviewer for finding the missing part in our DFT calculations. We performed additional DFT calculations on C-C coupling and realized that C-C coupling is enhanced in alkanethiolated slabs. The C-C bond formation, which is to obtain the key intermediate of COCHO*, is an essential step in generating C₂ compounds. Compared with pure Cu, C-C bond formation on the thiol-modified Cu surface is more thermodynamically and kinetically feasible, with a lower energy barrier of 0.28 eV to form COCHO*. This value increases to 0.35 eV on pure Cu, indicating that the modification by thiols on Cu could facilitate formation of the C-C bond, thereby leading to further production of C₂ compounds. We added these results to the manuscript on Page 4. The complete DFT pathway is presented in Figure S1.

“Next, we look at the C-C bond formation step to produce the COCHO* intermediate, an essential step in generating C₂ compounds. Compared with pure Cu, C-C bond formation on the thiol-modified Cu surface is thermodynamically and kinetically more favorable, with a lower transition barrier of 0.28 eV to form COCHO* (Supplementary Figure 1). This value increases to 0.35 eV on clean Cu, indicating that thiol-modified Cu kinetically favors C-C bond formation and thus, leads to more C₂ compounds.”

Q2. Generally, the catalytic activity of electrocatalysts is related to the number of active sites exposed on the catalyst surface, which can be reflected in the value of ECSAs. The authors mention that "thiol samples had 5x lower effective area than controls", since the ECSA values become smaller, won't it affect the final catalytic activity.

Many thanks for your valuable comment. Although the number of active sites is reduced significantly, each active site works more effectively, as predicted by our DFT calculations. However, if the coverage of thiols is too high, it would hamper the performance as observed for C₅S-CuNPs (see Table R10).

This clarification is added to page 9:

“Under CORR, the reaction onset potential (Figure 4a) of the thiol samples are lower by 100 mV compared to controls. To rule out surface area effects, we measured the electrochemically active surface area (ECSA) of the electrodes, finding that in fact the thiol samples had 4x lower effective area than controls.

This confirms that the per-site activity of the alkanethiolated samples is indeed higher than that of CuNPs based on the onset potential.”

Table R10. Electrochemical performance table of different alkanethiols at 100 mA/cm².

Catalyst	H ₂ FE (%)	C ₂ H ₄ FE (%)	CH ₄ FE (%)	C ₂ H ₅ OH FE (%)	C ₃ H ₇ OH FE (%)	CH ₃ COO ⁻ FE (%)
----------	-----------------------	--------------------------------------	------------------------	---	---	---

C ₂ S-Cu NPs	27.3±4.2	22.9±0.9	0.0±0.0	16.9±2.0	3.0±0.7	24.1±2.2
C ₅ S-Cu NPs	7.1±1.2	24.3±1.3	0.9±0.2	10.2±1.4	3.3±1.3	40.8±3.6
C ₁₂ S-Cu NPs	7.6±1.6	18.7±2.0	0.0±0.0	12.6±1.7	12.2±1.9	45.4±3.7
C ₁₈ S-Cu NPs	5.6±1.5	19.2±2.5	0.0±0.0	13.4±1.3	8.2±0.9	46.1±3.2
C ₂₂ S-Cu NPs	5.0±1.2	26.5±2.8	0.0±0.0	13.0±2.2	4.3±1.1	37.7±3.0
Cu NPs	39.7±2.3	10.8±1.2	0.0±0.0	7.4±1.8	5.4±0.8	13.5±1.9
Cu NPs+Nafion	9.4±1.1	37.8±2.3	0.0±0.0	13.6±1.5	15.3±2.1	14.4±1.3

Q3: The authors observed by STEM images that the surface of the sample was covered with a monolayer of thiols. Since it can be seen on STEM images, it means that the organic ligands on the surface reached a certain concentration, and such a high concentration of coverage may be the reason for the small ECSA value. In addition, the organic ligand contains 18 carbon atoms, wouldn't the high concentration of long alkyl chain coverage hinder the adsorption of reactants on the catalyst surface and thus have an effect on the activity of catalytic CO to produce acetate?

We agree with the reviewer that the surface of the sample was covered with a monolayer of thiols based on the STEM image. This means that the organic ligands on the surface reached a certain concentration, and such a high concentration of coverage may be the reason for the small ECSA value. We realized, in general, as we increase the carbon chain length, we increase the overpotential, resulting in the kinetics of the reaction becoming slower, possibly due to diffusion limitation of reactants. However, using shorter chain thiols would increase their packing as well, leading to a reduction of reactant diffusion at high current densities (>100mA/cm²). We found the optimum carbon chain length to be C₁₂ and C₁₈, which have both a suitable overpotential, and could reach high current densities.

Moreover, the alkyl carbon chain is necessary to dissolve CO and makes it available for the CORR. (please refer to Reviewer 2 comment 9).

We added a few notes explaining these points on pages 10 and 11:

“To better understand the alkane-thiol coverage, the ECSA was measured for all the RS-CuNPs samples (Supplementary Table 6). We posit there to be a direct relation between the ECSA and alkanethiol coverage such that the alkanethiolated sites do not contribute to the ECSA measurement. Moreover, the alkane-thiols would block ions from reaching the surface and therefore reduce the ECSA further.

On the C₂S-CuNPs sample, the alkyl chain of only two carbons is not as effective in blocking the ions, evident from its higher ECSA (29.3 cm²_{ecsa}) compared to other RS-CuNPs. The ECSA of C₂S-CuNPs catalyst is almost half of bare CuNPs (62 cm²_{ecsa}). As shown in Supplementary Table 6, C₁₂S-CuNPs and C₁₈S-CuNPs have similar ECSA (16.5 and 15.5 cm²_{ecsa}, respectively) and C₅S-CuNPs has a slightly lower ECSA (8.3 cm²_{ecsa}), attributed to the denser packing of the smaller alkane-thiols (C₅ vs. C₁₂ and C₁₈). The C₂₂S-CuNPs had an ECSA of only 5 cm²_{ecsa}, due to the long alkane chains inhibiting ion diffusion to the surface.

We explored the effect of ligand loading, finding that higher loadings inhibit HER (Figure 4b); an effect that saturates, presumably after a near-monolayer coverage is achieved. Higher loadings of C₁₈SH render the surface more hydrophobic and reduce the ECSA, demonstrative of a high coverage of alkane-thiols (Supplementary Table 7). The ECSA reduction continues until a loading of 5 mg, with no significant change in ECSA at a higher loading of 15 mg (Supplementary Figure 19). It is worth mentioning that the initial loading is not the same as the loading during the electrochemical reaction as we expect the physisorbed alkanethiols to be removed in alkaline conditions.”

Q4: The authors concluded that the alkanethiol length has little effect on the performance, so why not use ligands with short carbon chains? Moreover, considering that the authors used C₂SH in the theoretical calculation, it is suggested that the experimental data of C₂SH should be added when comparing the effect of carbon chain length on catalytic performance, the results will be more convincing.

Thanks for pointing this out. We have performed more experiments with different thiol chain lengths length (from C₂ to C₂₂). For C₂₂ to C₆, at 100 mA/cm² (Table R10), we observed negligible effect on the product distribution. However, a measurable change in overpotential and ability to reach higher current densities was observed. Although C₂ promoted acetate production, possibly due to its high coverage, we observed a higher HER FE (~30%) at 100 mA/cm². Furthermore, we posit the C₂ to be too short to be effective in dissolving CO, therefore limiting CORR current densities. Despite this, HER FE as low as ~5% is achieved at 50 mA/cm² for C₂S-CuNPs. Moreover, C₂S-CuNPs showed a better FE for acetate compared to CuNPs. We discussed these points and data for C₂ on pages 10 and 11.

Q5: Make sure all abbreviations are written out in full the first time used. For example, C₁₈S-CuNPs.

We checked all the abbreviations and made sure all were written in full.

Q6: When the electrode potentials were converted to the RHE, the authors used a standard potential of 0.235 V for Ag/AgCl. However, in the authors' previous work, 0.197 V was used (ACS Energy Lett. 2021, 6, 79-84), and 0.197 V was used in most of the previously reported works. It is suggested that the authors give a reason for choosing 0.235V.

Thanks for the comment. We used the same potential as *Nature Catalysis* 2.12 (2019): 1124-1131. The reference electrode was stored in a saturated KCl solution, and I suspected the E⁰ of that reference was shifted to higher values. Later for this revision, I used a brand-new reference electrode with 1M KCl electrolyte (voltage is presented in table 4). We realized the voltage is about 40 mV higher (more positive) than the old data, which is roughly the same as the difference between 0.235 V and 0.197 V.

Furthermore, the same reference electrode was used for all the old experiments. Therefore, it can be used for comparison. We apologize for this, and we changed the Voltage for conversion to 0.197 V in the methods section.

Q7: Can alkanethiol be replaced by other ligands containing lone pairs? For example, aliphatic amine

We attempted to replace alkanethiol with oleylamine ligand with no success as amines did not remain attached to the copper during synthesis. The resulting catalyst was hydrophilic and performed similarly to bare copper (see Table R11).

Moreover, our new DFT calculation showed that alkane thiols are the only ligands that can be effective (refer to Reviewer 2 comment 1). In short, alkoxides and selenides were not as effective in interacting with CORR intermediates.

Table R11. The CORR performance comparison between oleylamine-CuNPs and CuNPs catalysts.

Catalyst	H ₂ FE (%)	Ethylene FE (%)	Methane FE (%)	Ethanol FE (%)	Propanol FE (%)	Acetate FE (%)
Oleylamine+Cu NPs	35.0%±5.5	8.6%±2.3	0.5%±0.2	10.0±2.2	2.1±0.8	16.0±2.3
Cu NPs	39.7±2.3	10.8±1.2	0.0±0.0	7.4±1.8	5.4±0.9	13.5±1.9

We revised the manuscript by adding a paragraph on page 12 and the results to supplementary Table 10.

“We examined whether a hydrophobic amine like oleylamine can behave similarly in CORR. However, the performance was similar to bare CuNPs suggesting that oleylamine did not remain attached to copper under our experimental conditions (Supplementary Table 10).”

Q8: It is recommended to add a performance comparison table.

The performance comparison table is presented in Supplementary Tables 3 and 8.

Q9: Mismatch between captions and images in Figure 2.

The mismatch between captions and images in Figure 2 is fixed.

Q10: There should be a discussion of the work at the end of the article.

The following discussion is added at the end of the article:

“Discussions:

Overall, we explored two aspects of alkanethiols: the impact of available sulfur lone pairs adjacent to the surface CORR intermediates and alkyl group length. Sulfur lone pairs were directly involved in the transformation of CORR intermediates and reduced the onset potential for CORR by ~100 mV. The Intermediates having empty π^* are the most affected by sulfur lone pairs and they are crucial for obtaining acetate. Such lateral interaction is uncommon in molecular catalysis since the geometry of a single metal site does not allow for effective interaction.

On the other hand, the alkyl groups assist in the formation of triple-phase boundary, and their size mainly affects the packing and diffusion. As a result of the low diffusion of CORR reactants due to excessive coverage or long alkyl chain (in the case of C₂₂), HER rises at current densities above 100 mA/cm². These catalysts exhibit relatively high acetate FE, indicating they follow the same

reaction mechanism. C₂S-CuNPs is a special case in which ethyl groups are insufficient to establish proper triple-phase boundaries. We conclude that choosing the proper interacting element (sulfur in our case) and optimum alkyl chain (C₁₈) is crucial for obtaining a high-performance acetate electrocatalyst.”

REVIEWER COMMENTS

Reviewer #1 (Remarks to the Author):

I am satisfied with the authors' responses. The authors provided additional evidence and I believe their work can now be accepted for publication.

Reviewer #2 (Remarks to the Author):

The quality of this manuscript improves after revision. However, key issues are still missing and some of my questions are not fully solved. Therefore, I still can not recommend the publish of this manuscript in Nature Communications. Comments are listed below:

1. Although this work exhibits good acetate selectivity at high current densities (an acetate FE of 70% at 400 mA/cm²) compared to some acetate generation work for CORR, catalyst design does not have outstanding novelty. Luo et al. developed a Cu(I)-benzimidazole coordination polymer electrocatalyst for efficient CO-to-acetate conversion and this catalyst enabled a 61% Faradaic efficiency toward acetate at a current density of 400 mA cm⁻² (Adv. Mater. 2023, 35, 2209567). In this manuscript, the ligand is simply replaced with thiol surface ligands while the acetate selectivity is only slightly improved. Furthermore, a previous work has demonstrated that excellent Cu-based catalysts exhibit better electrochemical reduction performance for acetate production in CORR. Ji et al. developed a CuPd electrocatalyst which can enable a high Faradaic efficiency of 70 ± 5% for CO-to-acetate electroreduction and a high acetate partial current density of 425 mA cm⁻² and the CuPd electrocatalyst demonstrated a 500 h CO-to-acetate conversion at 500 mA cm⁻² with a stable acetate Faradaic efficiency under membrane electrode assembly conditions (Nat. Catal. 2022, 5, 251-258). The author's catalyst does not represent competitive selectivity and stability.
2. For Q1 replied by the authors: Although the author maintained "under reduction, we expect all the Cu(II) and Cu(I) to convert to metallic copper for CuNPs" and "copper atoms in C18S-CuNPs have an oxidation state of 0 except the ones attached to thiol surface ligands". But no direct evidence is provided to certify this point. Since Cu(I) species would usually be generated during CO₂RR (such as Nat. Energy 2020, 5, 317; Nat. Commun. 2020, 11, 3525.), the valence conversion of Cu should be more carefully evaluated and experimental data should be provided.
3. For Q5 replied by the authors: The authors claimed that the peak related to (HOOC-CH₂)* on copper is red-shifted by almost 200 cm⁻¹ due to the sulfur lone interaction with the empty orbitals of -COOH, the peak located at ~1600 cm⁻¹ is redshifted to ~1400 cm⁻¹ and this is rather surprising. Can the sulfur lone interaction with the empty orbitals of -COOH cause such a significant redshift in peak position? In fact, the Raman spectrum of CuNPs shows that there appears to be a peak at 1400 cm⁻¹ from Figure R6. Do other thiol surface ligands have similar redshifts? Is the redshift wavenumber related to the alkane chain length of the thiol surface ligands?
4. Theoretical calculations for Cu functionalized with CH₃CH₂CH₂-, CH₃CH₂O- and CH₃CH₂Se- are performed to prove the beneficial of CH₃CH₂S-. However, the authors do not offer any experimental

evidence. Maybe $\text{CH}_3\text{CH}_2\text{CH}_2^-$ can not be stably functionalized on Cu, but other groups such as $\text{CH}_3\text{CH}_2\text{Se}^-$ should be tested.

5. The calculation model also need to be further optimized. According to Supplementary Table 3, C18S-CuNPs demonstrate the best catalytic performance for the production of acetate. Therefore, C18S-Cu is suggested to be used for theoretical calculation rather than C2S-Cu.

6. The authors suggest “the C5S-CuNPs exhibited a considerable limitation in CORR current density, with HER FE values spiking to > 60% at 200 mA cm⁻²”. But the electrochemical performance of C5S-CuNPs under different current densities are not demonstrated. Similar data are also missing for C2S-CuNPs and C22S-CuNPs. Besides, the authors declare that “on higher chain length samples such as C22S-CuNPs, an increase in HER is also observed at 100 mA/cm²”. But the FE of H₂ for C22S-CuNPs is only 5%, which is even lower than C18S-CuNPs and C12S-CuNPs. I do not think this result can be used to prove that “large alkyl chains limit the ion and reactant diffusion onto the surface, inhibiting CORR activity”.

7. It seems that the alkyl groups also affect the CORR performance of the catalysts. But the authors do not give an in-depth theoretical understanding. How would the different alkyl groups affect the formation of triple-phase boundary? Molecular dynamics simulation may need to be considered during theoretical calculations.

8. The method to evaluate the surface coverage of the ligand is too rough. The authors test CV in the non-faradaic region to evaluate ECSA and then use ECSA to quantify the coverage. But would alkanethiolated sites really not contribute to the capacitive behavior of the electrode? Are different alkanethiolated sites show similar non-faradaic behavior? More study should be provided to further illustrate the surface coverage of the different catalysts.

9. Besides, I can not agree with the authors opinion that the previous work (ACS Cent. Sci. 2017, 3, 1032) “achieved at low current densities (1.34 mA/cm²) in H-cell and cannot be extrapolated to industry-relevant current densities”. One of the reasons that this work achieves large current density is that flow cell or MEA is used. It is more like a technique issue rather than a scientific issue. It is unfair to compare the current density for different devices.

Reviewer #3 (Remarks to the Author):

The authors have satisfactory provided point by point explanation to the concerns raised and have corrected all ambiguities in the manuscript accordingly. Therefore, I recommend that this manuscript could be published without further revision.

Response to the reviewers' comments

We are grateful for the insightful and valuable comments provided by the reviewers. We have conducted new experiments and surveyed literature, the results of which are now reflected in the main manuscript and supplementary information. The additions and changes are highlighted in yellow in the revised manuscript package attached.

Reviewer #1 (Remarks to the Author):

I am satisfied with the authors' responses. The authors provided additional evidence and I believe their work can now be accepted for publication.

Reviewer #2 (Remarks to the Author):

The quality of this manuscript improves after revision. However, key issues are still missing and some of my questions are not fully solved. Therefore, I still can not recommend the publish of this manuscript in Nature Communications. Comments are listed below:

1. Although this work exhibits good acetate selectivity at high current densities (an acetate FE of 70% at 400 mA/cm²) compared to some acetate generation work for CORR, catalyst design does not have outstanding novelty. Luo et al. developed a Cu(I)-benzimidazole coordination polymer electrocatalyst for efficient CO-to-acetate conversion and this catalyst enabled a 61% Faradaic efficiency toward acetate at a current density of 400 mA cm⁻² (Adv. Mater. 2023, 35, 2209567). In this manuscript, the ligand is simply replaced with thiol surface ligands while the acetate selectivity is only slightly improved. Furthermore, a previous work has demonstrated that excellent Cu-based catalysts exhibit better electrochemical reduction performance for acetate production in CORR. Ji et al. developed a CuPd electrocatalyst which can enable a high Faradaic efficiency of 70 ± 5% for CO-to-acetate electroreduction and a high acetate partial current density of 425 mA cm⁻² and the CuPd electrocatalyst demonstrated a 500 h CO-to-acetate conversion at 500 mA cm⁻² with a stable acetate Faradaic efficiency under membrane electrode assembly conditions (Nat. Catal. 2022, 5, 251-258). The author's catalyst does not represent competitive selectivity and stability.

We have now referenced both works in the manuscript, and add the following sentence to the introduction: "Surface alloying and polymer coordination have been reported as strategies to enhance acetate selectivity^{7,8}, however those strategies only modulate the active site characteristics and rely on other means to control the reaction environment e.g. ionomers. Here we demonstrate a ligand modification strategy where the anchoring atom stabilizes acetate-specific intermediates electronically, while the ligand tail can be tailored to modify the local reaction environment."

2. For Q1 replied by the authors: Although the author maintained "under reduction, we expect all the Cu(II) and Cu(I) to convert to metallic copper for CuNPs" and "copper atoms in C18S-CuNPs have an oxidation state of 0 except the ones attached to thiol surface ligands". But no direct evidence is provided

to certify this point. Since Cu(I) species would usually be generated during CO₂RR (such as Nat. Energy 2020, 5, 317; Nat. Commun. 2020, 11, 3525.), the valence conversion of Cu should be more carefully evaluated and experimental data should be provided.

We have included the following sentence in the manuscript to clarify our conclusion from the operando XAS experiments: “Prior reports in CO₂RR have indicated the presence of Cu(I) species under pulsed conditions^{19,20}, however under constant current experiments and in CORR conditions where the buffering capacity of CO₂-carbonate is not present, we find that Cu NPs transform to their Cu(0) oxidation state. Moreover, results from DFT calculations (Supplementary Note 2, Supplementary Figure 29) find the Cu atoms attached to alkanethiols to have a positive charge. This is reflected in the shift in XANES curves of the C₁₈S-Cu NP compared to Cu NP (Figure R1).”

Figure R1: XANES spectra of C₁₈S-Cu NPs in operando and ex-situ conditions compared with Cu NPs measured at -1.7 V Vs. Ag/AgCl (not iR corrected).

3. For Q5 replied by the authors: The authors claimed that the peak related to (HOOC-CH₂)* on copper is red-shifted by almost 200 cm⁻¹ due to the sulfur lone interaction with the empty orbitals of -COOH, the peak located at ~1600 cm⁻¹ is redshifted to ~1400 cm⁻¹ and this is rather surprising. Can the sulfur lone interaction with the empty orbitals of -COOH cause such a significant redshift in peak position ?

In fact, the Raman spectrum of CuNPs shows that there appears to be a peak at 1400 cm⁻¹ from Figure R6. Do other thiol surface ligands have similar redshifts? Is the redshift wavenumber related to the alkane chain length of the thiol surface ligands?

Obtaining irrefutable evidence from Raman spectroscopy that the peak shift is directly attributable to the Cu-S is quite challenging. We have hence omitted claims of this peak shift being due to the Cu-S interaction with the CO* and rely on DFT for the remainder of the manuscript.

4. Theoretical calculations for Cu functionalized with CH₃CH₂CH₂-, CH₃CH₂O- and CH₃CH₂Se- are performed to prove the beneficial of CH₃CH₂S-. However, the authors do not offer any experimental evidence. Maybe CH₃CH₂CH₂- can not be stably functionalized on Cu, but other groups such as CH₃CH₂Se- should be tested.

Unfortunately, we have encountered difficulties in finding a reliable chemical supplier to obtain CH₃CH₂SeH, and we also have concerns regarding its toxicity. To minimize reader confusion, we have removed the CH₃CH₂Se-related results from the manuscript. It is important to note that this omission does not impact the overall conclusion of this work.

Regarding the experimental functionalization of the Cu surface with CH₃CH₂CH₂- and CH₃CH₂O-, we acknowledge that it poses significant challenges. To gain further insights into these processes, we believe that conducting theoretical calculations would be a suitable approach.

5. The calculation model also need to be further optimized. According to Supplementary Table 3, C18S-CuNPs demonstrate the best catalytic performance for the production of acetate. Therefore, C18S-Cu is suggested to be used for theoretical calculation rather than C2S-Cu.

We have added the following references and sentences to the manuscript to address the reviewer's comment:

“prior reports have shown that longer alkyl chains would have minimal influence on the calculated relative energies of intermediates⁷. Hence, we simulate a -SCH₂CH₃ ligand to show the effect of sulfur lone pair in our calculations.”

Similarly since our primary objective is to investigate the role of sulfur in alkanethiols in the interaction with CORR intermediates, utilizing C₂ thiolates as substitutes for C₁₈ thiolates in the DFT calculations preserves the effect of sulfur.

6. The authors suggest “the C5S-CuNPs exhibited a considerable limitation in CORR current density, with HER FE values spiking to > 60% at 200 mA cm⁻²”. But the electrochemical performance of C5S-CuNPs under different current densities are not demonstrated. Similar data are also missing for C2S-CuNPs and C22S-CuNPs. Besides, the authors declare that “on higher chain length samples such as C22S-CuNPs, an increase in HER is also observed at 100 mA/cm²”. But the FE of H₂ for C22S-CuNPs is only 5%, which is even lower than C18S-CuNPs and C12S-CuNPs. I do not think this result can be used to prove that “large alkyl chains limit the ion and reactant diffusion onto the surface, inhibiting CORR activity”.

We conducted electrochemical CORR tests on C₂S-CuNPs, C₅S-CuNPs, and C₂₂S-CuNPs at a current density of 200 mA/cm², and the summarized outcomes can be found in Table R1. We did not perform electrochemical measurements For C₂S-CuNPs, C₅S-CuNPs, and C₂₂S-CuNPs at current densities higher than 200 mA/cm² because of high HER and flooding issues that occur at high HER FEs. The findings indicate that C₁₂ and C₁₈ demonstrate optimal performance at high current densities, while C₂, C₅, and C₂₂ exhibit limitations at higher current densities, likely due to constraints arising from reactant diffusion.

In the previous revision, we unintentionally did not include the 200 mA/cm² data for C₂₂S-CuNPs and we mistakenly wrote 100 mA/cm². It is now fixed.

Table R1. Summary of Faradaic efficiencies of RS-CuNP catalysts at 200 mA/cm².

Catalyst	H ₂ FE (%)	C ₂ H ₄ FE (%)	CH ₄ FE (%)	C ₂ H ₅ OH FE (%)	C ₃ H ₇ OH FE (%)	CH ₃ COO ⁻ FE (%)
C ₂ S-Cu NPs	60.0±4.7	6.2±0.9	0.7±0.3	2.3±1.1	0.1±0.1	10.1±1.9
C ₅ S-Cu NPs	63.9±2.7	4.0±0.9	2.5±0.5	16.9±2.0	0.1±0.1	9.9±2.5
C ₁₂ S-Cu NPs	7.2±1.9	20.1±2.0	0.0±0.0	8.4±1.1	9.1±1.9	50.9±3.7
C ₁₈ S-Cu NPs	6.1±1.3	20.0±1.8	0.0±0.0	10.3±1.1	7±1.0	58.3±4.1
C ₂₂ S-Cu NPs	49.0±5.1	9.5±2.1	1.5±0.1	5.2±2.6	0.4±0.2	23.6±3.6

We revised the manuscript by including these results in Supplementary Table 4 and adding this to the manuscript: “On higher chain length samples such as C₂₂S-CuNPs, an increase in HER is also observed at 200 mA/cm²”.

7. It seems that the alkyl groups also affect the CORR performance of the catalysts. But the authors do not give an in-depth theoretical understanding. How would the different alkyl groups affect the formation of triple-phase boundary? Molecular dynamics simulation may need to be considered during theoretical calculations.

We revised the manuscript by including the following in the main text on page 9: “We understand that alkyl length should have significant impact on the product distribution since it affects the reactant diffusion and coverage^{7,14,23}.”

8. The method to evaluate the surface coverage of the ligand is too rough. The authors test CV in the non-faradaic region to evaluate ECSA and then use ECSA to quantify the coverage. But would alkanethiolated sites really not contribute to the capacitive behavior of the electrode? Are different alkanethiolated sites show similar non-faradaic behavior? More study should be provided to further illustrate the surface coverage of the different catalysts.

According to our observations from ECSA measurements, we found that the coordination of alkanethiols leads to a reduction in surface capacitance (refer to Supplementary Table 6). This phenomenon is consistent with previous findings where the gold surface was thiolated using thiols. Alkane thiols are commonly employed to passivate electrode surfaces in biosensors.¹

¹ Shaver, Alexander, Samuel D. Curtis, and Netzahualcoyotl Arroyo-Curras. "Alkanethiol monolayer end groups affect the long-term operational stability and signaling of electrochemical, aptamer-based sensors in biological fluids." *ACS applied materials & interfaces* 12, no. 9 (2020): 11214-11223.

We acknowledge the reviewer's point regarding ECSA as a rough estimate of surface coverage. To further evaluate the coverage of alkanethiolated copper surfaces, we conducted ICP (inductively coupled plasma) measurements on C₂S-CuNPs to C₂₂-SCuNPs. As we previously mentioned and anticipated, we expected that smaller thiols would exhibit higher coverage on the copper surface, and indeed, this was confirmed through the S to Cu ratio observed in the ICP measurements (Figure R2). We observed a trend indicating that RS-CuNPs with shorter alkane chains displayed a higher S to Cu ratio, while catalysts with longer alkane chains exhibited a lower S to Cu ratio. This trend was roughly observed in the ECSA measurements as well (refer to Supplementary Table 6).

We revised the manuscript by including Supplementary Figure 25 and adding the following to the text on page 10 "To support these findings, we conducted Inductively Coupled Plasma (ICP) measurements. Our ICP results revealed RS-CuNPs with shorter chain alkane thiols have higher S:Cu ratio than RS-CuNPs with longer alkanethiols (Supplementary Figure 22).".

Figure R2: The ICP-OES results indicate a reverse relation between the coverage and alkylthiol length in RS-CuNPs catalysts.

9. Besides, I can not agree with the authors opinion that the previous work (ACS Cent. Sci. 2017, 3, 1032) "achieved at low current densities (1.34 mA/cm²) in H-cell and cannot be extrapolated to industry-

relevant current densities". One of the reasons that this work achieves large current density is that flow cell or MEA is used. It is more like a technique issue rather than a scientific issue. It is unfair to compare the current density for different devices.

We revised the manuscript by citing this paper and adding the following sentence to page 3: "Thiol ligands has been previously used in CORR to affix supramolecular cages to the copper surface".

Reviewer #3 (Remarks to the Author):

The authors have satisfactory provided point by point explanation to the concerns raised and have corrected all ambiguities in the manuscript accordingly. Therefore, I recommend that this manuscript could be published without further revision.

REVIEWER COMMENTS

Reviewer #2 (Remarks to the Author):

The quality of this manuscript does not show significant improvement after this revision. Further revision is still needed before considering its publication in Nature Communications. Comments are listed below:

1. For Q2 replied by the authors: The valence states of Cu in Figure R1 should be quantified to further verify the data in Supplementary Figure 29.
2. For Q3 replied by the authors: The authors omit the Raman discussion. But the question still been not solved. Is there any experimental evidence to prove the conversion of (HOC-CH₂O)* intermediate to (HOOC-CH₂)*? For a high quality journal such as Nature Communications, simple calculation evidence is not enough.
3. For Q4 replied by the authors: Similar question remains. Experimental evidence should be provided by the authors in addition to theoretical calculation results to further prove the beneficial of thiol.
4. For Q5-7 replied by the authors: It is obvious that the alkyl groups affect the CORR performance of the catalysts. If “longer alkyl chains would have minimal influence on the calculated relative energies of intermediates”, why would they demonstrate such different catalytic performance? DFT calculation should be applied on the optimized sample rather than C2S-Cu. Moreover, if the authors think that “alkyl length should have significant impact on the product distribution since it affects the reactant diffusion and coverage”. Calculations such as molecular dynamics simulation should be carried out to prove it.
5. For Q8 replied by the authors: The surface thiol coverage of the different catalysts should be quantified and discussed.

Response to the reviewers' comments

We are grateful for further feedback from reviewer 2. We have conducted new experiments and improved the presentation of prior literature. The additions and changes are highlighted in yellow in the revised manuscript package attached.

Reviewer #2 (Remarks to the Author):

The quality of this manuscript does not show significant improvement after this revision. Further revision is still needed before considering its publication in Nature Communications. Comments are listed below:

1. For Q2 replied by the authors: The valence states of Cu in Figure R1 should be quantified to further verify the data in Supplementary Figure 29.

In light of this recommendation, we now quantified the valence states of Cu based on experimental data. Taking into account the XANES spectra¹ of cuprous oxide, we now estimate the charge on R-SCu NPs and CuNPs under reduction conditions, reporting +0.2 and 0, respectively. We now detail the methodology in the revised version.

We added the following to Supplementary Note 2: “We estimated the average oxidation state of copper atoms under different conditions (Supplementary Figure 28b). The oxidation state of Cu in the R-SCu NPs and CuNP samples under reduction conditions are estimated to be +0.2 and 0 respectively, based on the linear relationship between copper K-edge and oxidation state²”.

It is worth mentioning that Supplementary Figure 29, generated through DFT analysis, suggests that the surface copper atoms in R-SCu NPs exhibit partial positive charges. Theoretically and based on previous studies, the copper should have an oxidation state of +1^{3,4}. The presence of a +1 oxidation state in copper atoms bonded to RS- ligands is supported by **three** experimental techniques: EELS analysis of Cu L-edge (Figure 2c and Supplementary Figure 12), Cu L-edge TEY spectra of C18S-Cu NPs (Figure 2b), and CuLMM XPS data in Supplementary Figure 10.

Figure R1: Estimated average oxidation state of catalysts under *in situ* and *ex situ* conditions. The position of Cu₂O K-edge is based on the data previously published¹.

2.For Q3 replied by the authors: The authors omit the Raman discussion. But the question still been not solved. Is there any experimental evidence to prove the conversion of (HOC-CH₂O)* intermediate to (HOOC-CH₂)*? For a high quality journal such as Nature Communications, simple calculation evidence is not enough.

As the intermediates in question appear after the RDS, their coverage is very low and deterministic evidence from Raman spectroscopy for their observation cannot be provided. With this in mind, we have modified the manuscript text to clarify that the proposed change in mechanism is purely theoretical and we only speculate regarding its physical consequences.

In the revised text, we now explicitly emphasize the referee's point, which we accept: We now write on page 8, "It will be of interest to seek experimental evidence of the conversion of (HOC-CH₂O)* intermediate to (HOOC-CH₂)* and to observe, experimentally, the lone pair interaction at this step. In the present study, we were not able to witness this directly, something we attribute to the fact that the conversion of (HOC-CH₂O)* intermediate to (HOOC-CH₂)* happens after the RDS, and thus its coverage is low."

We note that the conversion of ((HOC-CH₂O)* intermediate to (HOOC-CH₂)* has been reported previously^{5, 6}. However, the previous report described a three membered ring intermediate instead of (HOOC-CH₂)*. In our DFT calculation, we started with the three membered ring, but it converts to (HOOC-CH₂)* and is not stable. Whether it is a three membered ring (epoxide-like) or (HOOC-CH₂)*, both are susceptible to interaction with the lone pairs on sulfur, and we believe this does not change the overall message of our work.

3. For Q4 replied by the authors: Similar question remains. Experimental evidence should be provided by the authors in addition to theoretical calculation results to further prove the beneficial of thiol.

In the revised work, we now summarize the benefits of alkane thiols, on page 14: “Using alkanethiols bring four benefits overall: low overpotentials due to sulfur lone pair interaction, high acetate FEs and, low HER, and enhanced operating stability.”

We have systemically studied the effect of thiol functional groups by changing their alkane chain length and inclusion of other functional groups such as -OH. We then propose, with use of theoretical calculations, electrochemical measurements and spectroscopy, a mechanism by which acetate formation has been enhanced on Cu NPs decorated with thiol ligands. The experimental data is presented in Figure 4 and Supplementary Figures 3-11 illustrates the advantages of alkanethiol ligands. The contain more than 70 experiments showing the beneficiary effects of a thiol surface modification for acetate production on Cu NP.

4. For Q5-7 replied by the authors: It is obvious that the alkyl groups affect the CORR performance of the catalysts. If “longer alkyl chains would have minimal influence on the calculated relative energies of intermediates”, why would they demonstrate such different catalytic performance? DFT calculation should be applied on the optimized sample rather than C₂S-Cu. Moreover, if the authors think that “alkyl length should have significant impact on the product distribution since it affects the reactant diffusion and coverage”. Calculations such as molecular dynamics simulation should be carried out to prove it.

In the revised work, we seek now to better explain this point. We now write on page 11, “In view of the near-100% coverage of short chain C₂S-CuNPs, C-C coupling is difficult, and high HER efficiencies are the result.” We also now write on page 14 “These catalysts exhibit relatively high acetate FE, indicating they follow the same reaction mechanism. C₂S-CuNPs is a special case in which ethyl groups are insufficient to establish proper triple-phase boundaries and also have a high thiol coverage, leading to high HER.” We have also sought to improve the discussion of page 4 and performed additional DFT calculations on a longer alkyl chain thiol (Figure R2), now writing: “To examine the impact of alkane chain length on this step, we also performed -SCH₂CH₂CH₃; in comparison with CH₃CH₂S-Cu slab, the relative energy difference for this step over CH₃CH₂CH₂S-Cu slab is minimal (+0.05 eV) in the calculated reaction pathway (Supplementary Figure 1a and b). This emerges since methylene groups lack functionality and are distant from the catalytic site, and thus have little impact on reaction mechanism. However, the alkyl chain could affect the coverage and diffusion, a topic further discussed below.”

Regarding the suggestion to perform molecular dynamics simulations, we believe this would go beyond the scope of our current work. Future works can include detailed MD simulations that study diffusion of various species through a thiol layer on Cu NP and the corresponding effect on electrocatalytic performance.

Figure R2. Free energy diagram of the limiting step from (OC-CHO)* to (OC-CH₂O)* over (a) C₃S-Cu, and (b) C₂S-Cu.

5. For Q8 replied by the authors: The surface thiol coverage of the different catalysts should be quantified and discussed.

We now write in the methodology section:

$$V = \frac{4}{3}\pi r^3 = 8.1 \times 10^{-18} \text{ cm}^3$$

$$d_{\text{Cu}} = 8.96 \text{ g/cm}^3$$

$$m_{\text{Cu}} = V \times d_{\text{Cu}} = 7.3 \times 10^{-17} \text{ g}$$

$$MW_{\text{Cu}} = 63.55 \text{ g/mol}$$

$$n_{\text{Cu, bulk}} = 1.15 \times 10^{-18} \text{ mol} \times 6.022 \times 10^{23} \text{ atom/mol} = 6.9 \times 10^5 \text{ atoms}$$

$$A = 4\pi r^2 = 2 \times 10^{-11} \text{ cm}^2$$

$$r_{\text{Cu}} = 1.28 \text{ \AA}$$

$$n_{\text{Cu, surface}} = \frac{2 \times 10^{-11} \text{ cm}^2}{5.1 \times 10^{-16} \text{ cm}^2/\text{atom}} = 3.9 \times 10^4 \text{ atoms}$$

$$\frac{n_{\text{Cu, surface}}}{n_{\text{Cu, bulk}}} = 5.7\%$$

where V , m_{Cu} and n are volume, mass of Cu and number of atoms, respectively. Based on the percentage of surface atoms, we calculated the coverage (S to surface copper ratio) and plotted in Supplementary Figure 22 and Figure R3.

The data indicates that C₂S-CuNPs exhibit nearly complete coverage, which could account for the high HER percentages and low CORR efficiency, as most of the copper active sites are bound by thiol ligands.

We have updated the manuscript to incorporate these calculations as shown in Supplementary Figure 22. We now write on Page 11, “We also estimated the coverage of thiol ligands based on ICP results and calculating the percentage of surface copper atoms in 25 nm copper nanoparticles. We estimated the coverage of thiols for C₂, C₅, C₁₂, C₁₈ and C₂₂S-CuNPs as 85, 53, 47, 20 and 16% respectively (Supplementary Figure 22). This indicates that, to achieve high C-C coupling and acetate production at high

current densities, we require fewer than half of copper sites to be unligated. In view of the near-100% coverage of short chain C₂S-CuNPs, C-C coupling is difficult, and high HER efficiencies are the result.”

Figure R3: The ICP-OES results indicate a reverse relation between the coverage (S to Surface copper ratio) and alkythiol length in RS-CuNPs catalysts.

References:

1. Zelinka, S. L.; Kirker, G. T.; Sterbinsky, G. E.; Bourne, K. J. *Plos one* **2022**, 17, (1), e0263073.
2. Zhou, Y.; Che, F.; Liu, M.; Zou, C.; Liang, Z.; De Luna, P.; Yuan, H.; Li, J.; Wang, Z.; Xie, H. *Nature chemistry* **2018**, 10, (9), 974-980.
3. Keller, H.; Simak, P.; Schrepp, W.; Dembowski, J. *Thin Solid Films* **1994**, 244, (1-2), 799-805.
4. Ron, H.; Cohen, H.; Matlis, S.; Rappaport, M.; Rubinstein, I. *The Journal of Physical Chemistry B* **1998**, 102, (49), 9861-9869.
5. Garza, A. J.; Bell, A. T.; Head-Gordon, M. *Acs Catalysis* **2018**, 8, (2), 1490-1499.
6. Nitopi, S.; Bertheussen, E.; Scott, S. B.; Liu, X.; Engstfeld, A. K.; Horch, S.; Seger, B.; Stephens, I. E.; Chan, K.; Hahn, C. *Chemical reviews* **2019**, 119, (12), 7610-7672.

REVIEWER COMMENTS

Reviewer #2 (Remarks to the Author):

Some of the key issues are still not been fully addressed in the revised manuscript. For Q4, at least C18S-Cu rather than C3S-Cu should be calculated and compared with C2S-Cu to further prove the length of the alkyl chains have minimal influence on the calculated relative energies of intermediates. Moreover, the authors suggest they explore “two aspects of alkanethiols: the impact of available sulfur lone pairs adjacent to the surface CORR intermediates and alkyl group length”. But only the effect of sulfur lone pair is clearly revealed. The mechanism discussion on the effect of alkyl group length is full of hypothesis. In-depth theoretical evidence should be provided to certify this point. Only after that can this manuscript be accepted.

We are grateful for further feedback from reviewer 2. We have conducted new DFT calculations and improved the presentation of prior literature. The additions and changes are highlighted in yellow in the revised manuscript package attached.

Reviewer 2

Comment1: Some of the key issues are still not been fully addressed in the revised manuscript. For Q4, at least C18S-Cu rather than C3S-Cu should be calculated and compared with C2S-Cu to further prove the length of the alkyl chains have minimal influence on the calculated relative energies of intermediates.

Response:

Overall we have sought to make claims throughout whose scope is suitably-bounded. For example, we have made the title more observational/empirical, not insisting on the mechanism: “Ligand-modified Nanoparticle Surfaces Influence CO Electroreduction Selectivity.” Additionally, performing the DFT calculation on C₁₈S-Cu adds complexity but not necessarily a new insight on the reaction mechanism. The inclusion of an additional 10 carbon atoms and 20 hydrogen atoms to study C₁₈ would necessitate an excessively large vacuum in the z-direction of the simulation box and a progressively huge computational cost. To investigate the role of the alkyl chain and address the reviewer’s concern, we systematically investigated larger alkyl chain up to C₅. As can be seen below the results show comparable thermodynamic differences between the two critical steps in the free energy diagram, indicating that the length of the alkyl chain has no effect on our previous conclusions (Figures R1 and R2).

We included the above discussion in the revised manuscript on page 4:

“Prior reports have shown that increasing the alkyl chain length further has minimal influence on the calculated relative energies of intermediates¹⁰. We performed DFT calculations on -S(CH₂)₂CH₃ to -S(CH₂)₄CH₃ and found that – in comparison with the case of the CH₃CH₂S-Cu slab – the same rate-limiting step is increased by a modest 0.05 eV (Supplementary Figure 1a-d). This occurs since methylene groups lack functionality and are distant from the catalytic site, and thus have little impact on reaction mechanism. However, the alkyl chain could affect the coverage, a topic further discussed below.”

Figure R1. Free energy diagram depicting the limiting step; $(\text{OC-CHO})^*$ to $(\text{OC-CH}_2\text{O})^*$ over (a) $\text{C}_2\text{S-Cu}$, (b) $\text{C}_3\text{S-Cu}$, (c) $\text{C}_4\text{S-Cu}$, and (d) $\text{C}_5\text{S-Cu}$.

Figure R2. Free energy change of the limiting step ($(\text{OC-CHO})^*$ to $(\text{OC-CH}_2\text{O})^*$) as a function of the number of carbon atoms in alkyl chains attached to Cu surface. The comparable thermodynamic differences (maximum difference is 0.05 eV) indicates that the length of the alkyl chain has no effect on the earlier conclusions.

Comment2: Moreover, the authors suggest they explore “two aspects of alkanethiols: the impact of available sulfur lone pairs adjacent to the surface CORR intermediates and alkyl group length”. But only the effect of sulfur lone pair is clearly revealed. The mechanism discussion on the effect of alkyl group length is full of hypothesis. In-depth theoretical evidence should be provided to certify this point. Only after that can this manuscript be accepted.

Response:

We now write in the Discussion section:

“We identify a number of opportunities for future work informed by the findings herein. DFT studies would beneficially explore the full range of alkyl chain length, including to those values that are optimal experimentally, C₁₂-C₁₈. Chain length could also influence the diffusion of reactants, something that could be studied using molecular dynamics.”